# EXPLOITING REFLECTIONAL SYMMETRY IN HETEROGENEOUS MORL

## ABSTRACT

This work studies heterogeneous Multi-Objective Reinforcement Learning (MORL), where objectives exhibit considerable discrepancies in, amongst others, sparsity. The heterogeneity can cause dense objectives to overshadow sparse but long-term rewards, leading to sample inefficiency. To address this issue, we propose Parallel Reward Integration with reflectional Symmetry for heterogeneous MORL (PRISM), a novel algorithm that aligns reward channels and enforces reflectional symmetry as an inductive bias. We design ReSymNet, a theory-inspired model that aligns time frequency across objectives, leveraging residual blocks to gradually learn a 'scaled opportunity value' for accelerating exploration while maintaining the optimal policy. Based on the aligned reward objectives, we then propose SymReg, a reflectional equivariance regulariser to enforce reflectional symmetry in terms of agent mirroring. SymReg constrains the policy search to a reflection-equivariant subspace that is provably of reduced hypothesis complexity, thereby improving generalisability. Across MuJoCo benchmarks, PRISM consistently outperforms the baseline and oracle (with full dense rewards) in both Pareto coverage and distributional balance, achieving hypervolume gains of over 100% against the baseline and even up to 32% against the oracle. The code is at `https://anonymous.4open.science/r/reward_shaping-1CCB`.

## 1 INTRODUCTION

Reinforcement Learning (RL) has been approaching human-level capabilities in many decision-making tasks, such as playing Go (Silver et al., 2017), autonomous vehicles (Kiran et al., 2021), robotics (Tang et al., 2025a), and finance (Hambly et al., 2023). Multi-Objective Reinforcement Learning (MORL) extends this framework to handle multiple reward channels simultaneously, allowing agents to balance competing objectives efficiently (Liu et al., 2014; Hayes et al., 2022). For example, a self-driving car must constantly balance multiple goals, such as minimising travel time while maximising passenger safety and energy efficiency. Prioritising speed would compromise the safety objectives, introducing the need for flexible and robust policies that can optimise across diverse and sometimes conflicting goals.

This paper considers an important, yet premature, setting where reward channels exhibit considerable heterogeneity in facets such as sparsity. Dense objectives can overshadow their sparse and long-horizon counterparts, steering policies toward short-term gains, while neglecting the objectives that are harder to optimise but potentially more important. A straightforward approach is to employ reward shaping methods to align the reward channels. However, existing algorithms, such as intrinsic curiosity (Pathak et al., 2017; Aubret et al., 2019) and attention-based exploration (Wei et al., 2025), are developed for single-objective cases and have significant deficiencies: separately shaping individual objectives can distort the Pareto front and structures between objectives. This highlights a critical gap in the literature: MORL requires a reward shaping method that enables efficient integration of the parallel but heterogeneous reward signals, leveraging their intrinsic structure, in order to improve sample efficiency.

To this end, we propose Parallel Reward Integration with reflectional Symmetry for Multi-objective reinforcement learning (PRISM), a method that structurally shapes the reward channels and leverages the reflectional symmetry in agents in heterogeneous MORL problems. We design a Reward Symmetry Network (ReSymNet) that predicts the reward given the state of the system and any avail-

able performance indicators (e.g., dense rewards in this work). The available sparse rewards are used as supervised targets. In ReSymNet, residual blocks are employed to approximate the 'scaled opportunity value', which has been proven to help accelerate training, decrease the approximation error, while maintaining the optimal solution of the native reward signals (Laud, 2004). ReSymNet stacks residual blocks that progressively refine per-step predictions through additive corrections, reconstructing dense reward signals. It aims at maintaining consistent optima with the original sparse objectives while ironing out the heterogeneity and enhancing performance. After proper training, our ReSymNet can be a plug-and-play technique, compatible with any off-the-shelf MORL algorithm in an iterative refinement cycle, where the agent observes the shaped rewards to improve its policy and the reward model observes better trajectories from the updated policy to improve the approximated reward function. To exploit the structural information across reward signals, we design a Symmetry Regulariser (SymReg) to enforce reflectional equivariance of the objectives, which provably reduces the hypothesis complexity. Intuitively, incorporating reflectional symmetry as an inductive bias allows an agent to generalise experience from one situation to its mirrored counterpart.

The complementary components of PRISM synergise as follows. Heterogeneous reward structures cause asymmetric policy learning that violates the agent's physical symmetry: when dense objectives provide immediate gradients while sparse objectives only signal at the end of an episode, the policy may overfit to the denser objectives in specific states, failing to respect reflectional symmetry. ReSymNet eliminates temporal heterogeneity by aligning objectives to the same frequency, whereas SymReg enforces reflectional symmetry by preventing asymmetric learning dynamics.

We prove that PRISM constrains the policy search into a subspace of reflection-equivariant policies. This subspace is a projection of the original policy space, induced by the reflectional symmetry operator, provably of reduced hypothesis complexity, measured by covering number (Zhou, 2002) and Rademacher complexity (Bartlett & Mendelson, 2002). This reduced complexity is further translated to improved generalisation guarantees. In practice, this means that by encouraging policies to respect natural symmetries, the agent effectively searches over a smaller, more structured hypothesis space, reducing overfitting and improving sample efficiency. We further extend this analysis to the approximately reflection-equivariant cases, where PRISM does not necessarily converge to the reflection-equivariant subspace exactly, showing that policies in this more realistic setting inherit similarly improved generalisability.

We conduct extensive experiments on the MuJoCo MORL environments (Todorov et al., 2012; Felten et al., 2023), using Concave-Augmented Pareto Q-learning (CAPQL) (Lu et al., 2023) as the backbone for PRISM. Sparse rewards are constructed by releasing cumulative rewards at the end of an episode. PRISM achieves hypervolume gains of over 100% against the baseline operating directly on sparse signals, and even up to 32% over the oracle (full dense rewards), indicating a substantially improved Pareto front coverage. These gains are echoed in distributional metrics, confirming that PRISM learns a set of policies that are also better balanced and more robust. Comprehensive ablation studies further confirm that the design of ReSymNet and the inclusion of SymReg are both critical.

## 2 RELATED WORK

**Multi-Objective Reinforcement Learning.** MORL algorithms typically fall into three categories: (1) single-policy methods that optimise user-specified scalarisations (Moffaert et al., 2013; Lu et al., 2023; Hayes et al., 2022); (2) multi-policy methods that approximate the Pareto front by solving multiple scalarisations or training policies in parallel (Roijers et al., 2015; Van Moffaert & Nowé, 2014; Reymond & Nowé, 2019; Lautenbacher et al., 2025); and (3) meta-policy and single universal policy methods that learn adaptable policies given some preferences (Chen et al., 2019; Yang et al., 2019; Basaklar et al., 2023; Mu et al., 2025; Liu et al., 2025). While these works have advanced Pareto-optimal learning, less attention has been given to heterogeneity in reward structures.

**Reward Shaping.** A large volume of literature tackles sparse rewards through reward shaping. Potential-based shaping (Ng et al., 1999) ensures policy invariance but requires hand-crafted potentials. However, this method's reliance on a manually designed potential function proved limiting. Intrinsic motivation methods reward novelty or exploration (Pathak et al., 2017; Burda et al., 2019), while self-supervised methods predict extrinsic returns from trajectories (Memarian et al., 2021; Devidze et al., 2022; Holmes & Chi, 2025). Recent advances utilise statistical decomposition to

address sparsity (Gangwani et al., 2020; Ren et al., 2022), or capture complex reward dependencies using transformers (Tang et al., 2024; 2025b). These approaches improve sample efficiency in single-objective RL, but do not extend naturally to MORL, where heterogeneous sparsity and scale can distort learning dynamics and Pareto-optimal trade-offs.

**Reflectional Equivariance.** To incorporate reflectional symmetry, a possible method is data augmentation, which adds mirrored transitions to the replay buffer but doesn't guarantee a symmetric policy and increases data processing costs (Lin et al., 2020). Mondal et al. (2022) propose latent space learning that encourages a symmetric representation through specialised loss functions. Another line of research focuses on equivariant neural networks (van der Pol et al., 2020; Mondal et al., 2020; Wang et al., 2021). For example, Wang et al. (2022) design a stronger inductive bias via architecture-level symmetry, which hard-codes equivariance into the model for instantaneous generalisation. However, Park et al. (2025) show that strictly equivariant architectures can be too rigid for tasks where symmetries are approximate rather than perfect. Building on this insight, our framework helps overcome the limitations of strictly equivariant architectures through tunable flexibility whilst being model-agnostic.

## 3 Preliminaries

**Multi-Objective Markov Decision Process.** Formally, we define an MORL problem via the Multi-Objective Markov Decision Process (MOMDP) model, as a tuple $\mathcal{M} = (\mathcal{S}, \mathcal{A}, \mathcal{P}, \boldsymbol{r}, \gamma)$: an agent at state $s$ from a finite or continuous state space $\mathcal{S}$, taking action $a$ from a finite or continuous action space $\mathcal{A}$, moves herself according to a transition probability function $\mathcal{P} : \mathcal{S} \times \mathcal{A} \times \mathcal{S}' \rightarrow [0, 1]$, also denoted as $P(s'|s, a)$. The agent receives a reward via an $L$-dimensional vector-valued reward function $\boldsymbol{r} : \mathcal{S} \times \mathcal{A} \rightarrow \mathbb{R}^L$, where $L$ is the reward channel number, which decays by a discount factor $\gamma \in [0, 1)$. The goal in MORL is to find a policy $\pi : \mathcal{S} \rightarrow \mathcal{A}$ that optimises the expected cumulative vector return, defined as $\boldsymbol{J}(\pi) = \mathbb{E}_\pi \left[ \sum_{t=0}^{\infty} \gamma^t \boldsymbol{r}_t \right]$. This paper addresses episodic tasks, where each interaction sequence has a finite horizon and concludes when the agent reaches a terminal state, at which point the environment is reset. Episodes $\tau_i$ are i.i.d. draws from the behaviour distribution $\mathcal{D}$, which describes the probability of observing different possible trajectories under the policy being followed.

**Reward Sparsity.** Reward sparsity can be modelled as releasing the cumulative reward accumulated since the last non-zero reward with probability $p_{\text{rel}}$ at each timestep. When $p_{\text{rel}} = 0$, this reduces to the most extreme case: the agent receives rewards from dense channels $\mathcal{DC} = \{d_1, d_2, \ldots, d_D\}$ with observable rewards $r_t^{d_i}$ at every timestep, but the sparse channel is revealed only once at the end of the episode as $R_T^{sp} = \sum_{t=1}^{T} r_t^{sp}$. The central challenge is to recover instantaneous sparse rewards $r_t^{sp}$ for each $(s_t, a_t)$ using only the cumulative observation $R_T^{sp}$ and correlations with dense channels. Formally, given a trajectory $\tau = \{(s_1, a_1), \ldots, (s_T, a_T)\}$ with cumulative sparse reward $R^{sp}(\tau)$, the task is to infer $\boldsymbol{r}^{sp} = [r_1^{sp}, \ldots, r_T^{sp}]^\top$, where $r_t^{sp}$ is the sparse reward at timestep $t$, such that $\sum_{t=1}^{T} r_t^{sp} \approx R^{sp}(\tau)$. For $p_{\text{rel}} > 0$, an episode decomposes into sub-trajectories where the same formulation applies.

**Generalisability and Hypothesis Complexity.** A generalisation gap, at the episodic level, characterises the generalisability from a good empirical performance to its expected performance on new data (Wang et al., 2019). It depends on the hypothesis set's complexity, which is measured in this work by covering number (Zhou, 2002) and Rademacher complexity (Bartlett & Mendelson, 2002).

**Definition 1** ($l_{\infty,1}$ distance). *Let $\mathcal{X}$ be a feature space and $\mathcal{F}$ a space of functions from $\mathcal{X}$ to $\mathbb{R}^n$. The $l_{\infty,1}$-distance on the space $\mathcal{F}$ is defined as $l_{\infty,1}(f, g) = \max_{x \in \mathcal{X}} \left( \sum_{i=1}^{n} |f_i(x) - g_i(x)| \right)$.*

**Definition 2** (covering number). *The covering number, denoted $\mathcal{N}_{\infty,1}(\mathcal{F}, r)$, is the minimum number of balls of radius $r$ required to completely cover the function space $\mathcal{F}$ under the $l_{\infty,1}$-distance.*

**Definition 3** (Rademacher complexity). *Let $\mathcal{F}$ be a class of real-valued functions on a feature space $\mathcal{X}$, and let $\tau_1, \ldots, \tau_N$ be i.i.d. samples from a distribution over $\mathcal{X}$. The empirical Rademacher complexity of $\mathcal{F}$ is $\hat{\mathfrak{R}}_N(\mathcal{F}) = \mathbb{E}_\sigma[\sup_{f \in \mathcal{F}} \frac{1}{N} \sum_{i=1}^{N} \sigma_i f(\tau_i)]$, where $\sigma_1, \ldots, \sigma_N$ are independent Rademacher random variables taking values $\pm 1$ with equal probability. The Rademacher complexity of $\mathcal{F}$ is the expectation over the sample set.*

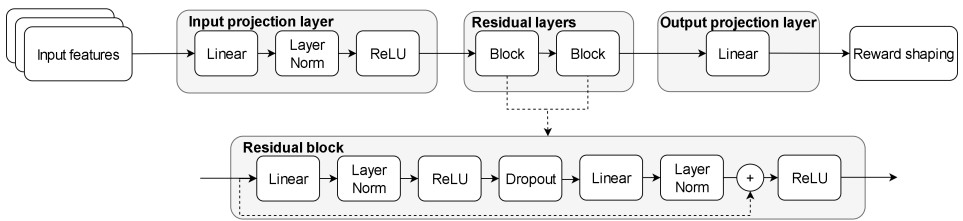

Figure 1: Overview of ReSymNet.

# 4 PARALLEL REWARD INTEGRATION WITH REFLECTIONAL SYMMETRY

This section introduces our algorithm PRISM.

## 4.1 RESYMNET: REWARD SYMMETRY NETWORK

To address the challenge of heterogeneous reward objectives, PRISM first transforms sparse rewards into dense, per-step signals. We frame this as a supervised learning problem, inspired by but distinct from inverse reinforcement learning, as we do not assume access to expert demonstrations (Ng & Russell, 2000; Arora & Doshi, 2021). The goal is to train a reward model, $\mathcal{R}_{\text{pred}}$, parametrised by $\psi$, that learns to map state-action pairs to individual extrinsic rewards.

We hope to train the reward shaping model on a dataset collected by executing a purely random policy, ensuring broad state-space coverage. For each timestep $t$, we construct a feature vector $\boldsymbol{h}_t = [s_t, a_t, \boldsymbol{r}_t^{\text{dense}}]$, where $s_t$ is the state, $a_t$ is the action, and $\boldsymbol{r}_{\text{dense},t}$ are the dense rewards obtained from taking action $a_t$ at state $s_t$, which crucially leverages the information from already-dense objectives to help predict the sparse ones. Figure 1 visualises the ResNet-like architecture.

**Remark 1.** *Residual connections in $\mathcal{R}_{pred}$ are inspired by the theory of scaled opportunity value (Laud, 2004), whose additive corrections preserve optimal policies, shorten the effective reward horizon, and improve local value approximation (see Appendix B).*

The network is optimised by minimising the mean squared error between the sum of its per-step predictions over a trajectory and the true cumulative sparse reward observed for that trajectory:

$$\mathcal{L}(\psi) = \sum_{\tau \in \mathcal{D}} \left( \sum_{t \in \tau} \mathcal{R}_{\text{pred}}(\boldsymbol{h}_t; \psi) - R^{sp}(\tau) \right)^2. \tag{1}$$

To ensure the learned reward function is robust and adapts to the agent's improving policy, we incorporate two techniques: (1) we train an ensemble of reward models to reduce variance and produce a more stable shaping signal, and (2) we employ iterative refinement: the reward model is periodically updated using new, on-policy data collected by the agent. This allows the reward model to correct for the initial distribution shift and remain accurate as the agent's behaviour evolves from random exploration to expert execution, as outlined in Algorithm 1 in Appendix B.

## 4.2 SYMREG: ENFORCING REFLECTIONAL EQUIVARIANCE

However, aligning reward frequencies alone is insufficient, as heterogeneous rewards cause the policy to learn asymmetrically across objectives, violating the agent's physical symmetry. To address this, we leverage reflectional symmetry as an inductive bias to prevent asymmetric policy learning. For example, for legged agents, flexing a leg is essentially the mirror image of extending it. Standard policies must learn both motions separately, wasting data. By encoding symmetry as an inductive bias, experience from one motion can be reused for its mirror, improving sample efficiency and robustness.

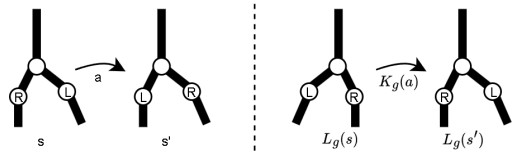

Figure 2: Reflectional symmetry in a two-legged agent. The left panel shows a transition from state $s$ to $s'$ under action $a$, whereas the right panel shows the reflected transition, where states and actions are transformed by $L_g$ and $K_g$, respectively.

We formalise this physical intuition using group theory, specifically the reflection group $G = \mathbb{Z}_2$. This group consists of two transformations: the identity and a negation/reflection operator, $g$. Let $\mathcal{S} \subseteq \mathbb{R}^{d_s}$ and $\mathcal{A} \subseteq \mathbb{R}^{d_a}$ denote the state and action spaces, respectively, where $d_s$ is the dimension of the state space and $d_a$ of the action space. We define index sets $I_{\text{asym}}^s \subset \{1, \ldots, d_s\}$ and $I_{\text{sym}}^s \subset \{1, \ldots, d_s\}$ such that $I_{\text{asym}}^s \cap I_{\text{sym}}^s = \emptyset$ and $I_{\text{asym}}^s \cup I_{\text{sym}}^s = \{1, \ldots, d_s\}$. This partitions the state vector as $s = (s_{\text{asym}}, s_{\text{sym}})$ where $s_{\text{asym}} = s_{I_{\text{asym}}^s}$ and $s_{\text{sym}} = s_{I_{\text{sym}}^s}$. We first partition the state vector $s$ into an asymmetric part, $s_{\text{asym}}$ (e.g., the torso's position), and a symmetric part, $s_{\text{sym}}$ (e.g., the leg's relative joint angles and velocities in Figure 2). The state transformation operator, $L_g : \mathcal{S} \to \mathcal{S}$, reflects the symmetric part of the state as follows: $L_g(s) = (s_{\text{asym}}, -s_{\text{sym}})$. Similarly, we define index sets $I_{\text{asym}}^a$ and $I_{\text{sym}}^a$ for the action space, and the action space is split up into an asymmetric part, $a_{\text{asym}}$, and a symmetric part, $a_{\text{sym}}$. The action transformation operator, $K_g : \mathcal{A} \to \mathcal{A}$, reflects the symmetric part of the action (e.g., the leg torques): $K_g(a) = (a_{\text{asym}}, -a_{\text{sym}})$.

The goal is to learn a policy, $\pi$, that is equivariant in terms of the aforementioned transformation. A policy $\pi$ is reflectional-equivariant if it satisfies the following condition for all states $s \in \mathcal{S}$: $\pi(L_g(s)) = K_g(\pi(s))$. This property means that the action for a reflected state is the same as the reflection of the action for the original state. To enforce this, we introduce a Symmetry Regulariser (SymReg) that explicitly penalises deviations from the desired symmetry property. During training, for each observation $s$, we compute both the standard policy output $\pi(a|s; \phi)$, parameterised by $\phi$, and the policy output for the reflected state $\pi(a|L_g(s); \phi)$. The equivariance loss is then defined as:

$$\mathcal{L}_{\text{eq}} = \mathbb{E}_{s \sim \mathcal{D}, a \sim \pi_\phi} \left[ \|\pi(a|L_g(s); \phi) - K_g(\pi(a|s; \phi))\|_1^2 \right].$$

SymReg measures the deviation between the policy's actual response to a reflected state and the expected reflected response. The total training objective combines the standard policy gradient loss, $J_\pi(\phi)$, with SymReg: $\mathcal{L}_{\text{total}} = J_\pi(\phi) + \lambda \mathcal{L}_{\text{eq}}$, where $\lambda$ is a hyperparameter controlling SymReg.

## 5 THEORETICAL ANALYSIS

This section presents theoretical guarantees of PRISM's generalisability. Let $\Pi$ be the full hypothesis space of policies represented by ReSymNet, $R(\pi; \tau)$ is the cumulative return for a single trajectory $\tau$ obtained following policy $\pi$.

**Remark 2.** *As the backbone of the whole method, the hypothesis complexity and generalisability of ReSymNet contribute significantly to the generalisability of the whole algorithm. Due to space limit, we present Theorem B.5 in the appendices for the covering number of ReSymNet's hypothesis space.*

The theory relies on these assumptions:

**Assumption 1** (bounded returns). *For all policies $\pi$ and trajectories $\tau$, $0 \leq R(\pi; \tau) \leq B$.*

**Assumption 2** (Lipschitz-continuous return). *There exists $L_R > 0$ such that for all $\pi, \tilde{\pi} \in \Pi$ and any trajectory $\tau$, $|R(\pi; \tau) - R(\tilde{\pi}; \tau)| \leq L_R d(\pi, \tilde{\pi})$, where $d(\pi, \tilde{\pi}) := \sup_{s \in \mathcal{S}} \|\pi(s) - \tilde{\pi}(s)\|_1$.*

**Assumption 3** (compact spaces). *The state space $\mathcal{S}$ and action space $\mathcal{A}$ are compact metric spaces.*

**Assumption 4** (bounded policy). *Policies $\pi \in \Pi$ have bounded inputs and weights.*

**Assumption 5** (episode sampling). *The behaviour distribution $\mathcal{D}$ has state marginal lower-bounded by $p_{\min} > 0$ on the state support of interest (finite-support or density lower-bound assumption).*

The Assumptions are reasonably mild. Bartlett et al. (2017) prove that feedforward ReLU are Lipschitz functions; since our policies are implemented as ReLU networks, this ensures bounded sensitivity of the policy outputs to perturbations. Assuming further that the return function is Lipschitz

in the policy outputs, it follows that returns are Lipschitz in the policies themselves, as stated in Assumption 2. Assumption 5 ensures that all relevant states are sufficiently sampled under the behaviour policy, which is, in practice, reasonable because policy exploration mechanisms prevent the policy from collapsing onto a subset of states.

## 5.1 GENERALISABILITY OF REFLECTION-EQUIVARIANT SUBSPACE

Let $G = \mathbb{Z}_2$ act on states and actions via $L_g, K_g$. An orbit-averaging operator $\mathcal{Q}(\pi)(s) = \frac{1}{2}\big(\pi(s) + K_g(\pi(L_g(s)))\big)$ maps any policy to a reflection-equivariant subspace (Qin et al., 2022). The regulariser $\mathcal{L}_{\text{eq}} = \mathbb{E}_s\|\pi(L_g(s)) - K_g(\pi(s))\|_1^2$ encourages convergence to the fixed-point subspace, defined as follows.

**Definition 4** (reflection-equivariant subspace). *We define reflection-equivariant subspace as* $\Pi_{\text{eq}} := \{\pi : \pi(L_g(s)) = K_g(\pi(s))\}$.

We prove that $\mathcal{Q}$ is reflectional equivariant, a projection, and that its image coincides with the set of equivariant policies in Lemmas C.4, C.5, and C.6 in Appendix C.3, respectively. Thus, $\mathcal{Q}$ is surjective onto $\Pi_{\text{eq}}$. To prove that the subspace $\Pi_{\text{eq}}$ is less complex, we show that the projection $\mathcal{Q}$ is non-expansive, which implies its image has a covering number no larger than the original space.

**Theorem 5.1.** *The space $\Pi_{eq}$ has a covering number less than or equal to that of $\Pi$. Let $\mathcal{N}_{\infty,1}(\mathcal{F}, r)$ be the covering number of a function space $\mathcal{F}$ under the $l_{\infty,1}$-distance. Then, $\mathcal{N}_{\infty,1}(\Pi_{eq}, r) \leq \mathcal{N}_{\infty,1}(\Pi, r)$.*

The $l_{\infty,1}$-distance between two policies $\pi_\phi$ and $\pi_\theta$ is $d(\pi_\phi, \pi_\theta) = \sup_s \|\pi_\phi(s) - \pi_\theta(s)\|_1$. The distance between their projections, $d(\mathcal{Q}(\pi_\phi), \mathcal{Q}(\pi_\theta))$, is no larger using the fact that $K_g$ is a norm-preserving isometry, $\|K_g(a)\|_1 = \|a\|_1$, and that $L_g$ is a bijection, which implies that the supremum over $s$ equals the supremum over $L_g(s)$. Hence $\mathcal{Q}$ is non-expansive, and a non-expansive surjective map cannot increase the covering number. Following Lemma C.6, $\mathcal{N}(\Pi_{\text{eq}}, r) \leq \mathcal{N}(\Pi, r)$. A detailed proof can be found in Appendix C.4.

The symmetrisation technique is fundamental in empirical process theory that reduces the problem of bounding uniform deviations to analysing Rademacher complexity (Bartlett & Mendelson, 2002).

**Corollary 5.2.** *For any class $\mathcal{F}$ of functions bounded in $[0, B]$, the expected supremum of empirical deviations satisfies:*

$$\mathbb{E}\left[\sup_{f \in \mathcal{F}}\left|\frac{1}{N}\sum_{i=1}^{N}(f(\tau_i) - \mathbb{E}[f])\right|\right] \leq 2\mathbb{E}[\mathfrak{R}_N(\mathcal{F})],$$

*where $\mathfrak{R}_N(\mathcal{F}) = \mathbb{E}_\sigma\left[\sup_{f \in \mathcal{F}} \frac{1}{N}\sum_{i=1}^{N}\sigma_i f(\tau_i)\right]$ is the Rademacher complexity and $\sigma_i$ are independent Rademacher random variables taking values $\pm 1$ with equal probability.*

This bound transforms the original centred empirical process into a symmetrised version that is often easier to analyse. We now prove a high-probability uniform generalisation bound over the reflection-equivariant subspace. A detailed proof can be found in Appendix C.5. We recognise that PRISM does not necessarily converge to it, which will be discussed in the following subsection.

**Theorem 5.3.** *With $\mathcal{R}_{\Pi_{\text{eq}}} = \{\tau \mapsto R(\pi; \tau) : \pi \in \Pi_{\text{eq}}\}$, fix any accuracy parameter $r \in (0, B)$ and confidence $\delta \in (0, 1)$. Then with probability at least $1 - \delta$,*

$$\sup_{\pi \in \Pi_{\text{eq}}}|J(\pi) - \hat{J}_N(\pi)| \leq C\left(\int_r^B \sqrt{\frac{\log \mathcal{N}_{\infty,1}(\mathcal{R}_{\Pi_{\text{eq}}}, \varepsilon)}{N}}d\varepsilon\right) + \frac{8r}{\sqrt{N}} + B\sqrt{\frac{\log(2/\delta)}{2N}},$$

*where $C$ is an absolute numeric constant, $J(\pi)$ is the population expected return and $\hat{J}_N(\pi) = \frac{1}{N}\sum_{i=1}^{N} R(\pi; \tau_i)$ is the empirical return on $N$ i.i.d. episodes $\tau_1, \ldots, \tau_N$.*

**Corollary 5.4.** *Under the same assumptions as Theorem 5.3, for any $r \in (0, B)$ and $\delta \in (0, 1)$, the upper bound in Theorem 5.3 for $\Pi_{\text{eq}}$ is at most the same bound obtained by replacing $\Pi_{\text{eq}}$ with $\Pi$. By Lemma C.8, the return-class covering numbers can be bounded by those of the policy class with radius scaled by $1/L_R$. Mathematically, following Theorem 5.1, for every $\varepsilon > 0$,*

$$\log \mathcal{N}_{\infty,1}\big(\Pi_{eq}, \varepsilon/L_R\big) \leq \log \mathcal{N}_{\infty,1}\big(\Pi, \varepsilon/L_R\big), \tag{2}$$

*hence the upper bound in Theorem 5.3 is no larger when evaluated on $\Pi_{\text{eq}}$.*

The equivariance regulariser projects policies onto a smaller fixed-point subspace $\Pi_{eq}$, which provably has covering numbers no larger than $\Pi$. The return class inherits this reduction via the Lipschitz map, so the Dudley entropy integral for $\Pi_{eq}$ is bounded by that of $\Pi$. As a consequence, the upper bound on the generalisation gap is no larger for $\Pi_{eq}$ compared to $\Pi$.

## 5.2 GENERALISABILITY OF PRISM

We now study the generalisability of PRISM, which does not necessarily converge to the reflection-equivariant subspace exactly. Rather, PRISM might converge to an approximately reflection-equivariant class. Using the orbit averaging $\mathcal{Q}$, we quantify this effect below.

**Definition 5** (approximately reflection-equivariant class). *Approximately reflection-equivariant class is defined as* $\Pi_{approx}(\varepsilon_{eq}) := \{\pi \in \Pi : \mathcal{L}_{eq} \leq \varepsilon_{eq}\}$.

**Theorem 5.5.** *Let* $\xi := \frac{1}{2}\sqrt{\varepsilon_{eq}/p_{\min}}$. *Then for every policy* $\pi \in \Pi$,

$$|J(\pi) - J(Q(\pi))| \leq L_R \cdot d(\pi, Q(\pi)) \leq L_R\xi. \tag{3}$$

*Then every* $\pi \in \Pi_{approx}(\varepsilon_{eq})$ *lies in the sup-ball of radius* $\xi$ *around* $\Pi_{eq}$. *Consequently, for any target covering radius* $r > \xi$, *we have:*

$$\mathcal{N}_{\infty,1}\big(\Pi_{approx}(\varepsilon_{eq}), r\big) \leq \mathcal{N}_{\infty,1}\big(\Pi_{eq}, r - \xi\big). \tag{4}$$

By Lipschitzness of returns, the expected return of a policy and its projection differ by at most $L_R d(\pi, Q(\pi))$. The mismatch $\Delta_\pi$ controls this distance, and Lemma C.10 bounds its supremum by $\xi$, giving the first inequality. Geometrically, $\Pi_{approx}(\varepsilon_{eq})$ is contained in a $\xi$-tube around $\Pi_{eq}$. Hence any $(r - \xi)$-cover of $\Pi_{eq}$ yields an $r$-cover of $\Pi_{approx}(\varepsilon_{eq})$, proving the covering-number relation (see Appendix C.6 for a detailed proof).

**Theorem 5.6.** *With* $\mathcal{R}_{\Pi_{eq}} = \{\tau \mapsto R(\pi; \tau) : \pi \in \Pi_{eq}\}$, *fix any accuracy parameter* $r \in (0, B)$ *and confidence* $\delta \in (0, 1)$. *Then with probability at least* $1 - \delta$,

$$\sup_{\pi \in \Pi_{approx}(\varepsilon_{eq})} |J(\pi) - \hat{J}_N(\pi)| \leq C\left(\int_r^B \sqrt{\frac{\log \mathcal{N}_{\infty,1}(\mathcal{R}_{\Pi_{eq}}, \varepsilon)}{N}}d\varepsilon\right) + \frac{8r}{\sqrt{N}} + B\sqrt{\frac{\log(2/\delta)}{2N}} + 2L_R\xi.$$

For $\pi \in \Pi_{approx}(\varepsilon_{eq})$, decompose the generalisation error relative to its projection $Q(\pi) \in \Pi_{eq}$. The difference in population returns $|J(\pi) - J(Q(\pi))|$ and in empirical returns $|\hat{J}_N(\pi) - \hat{J}_N(Q(\pi))|$ are both bounded by $L_R\xi$ (Theorem 5.5). The middle term $|J(Q(\pi)) - \hat{J}_N(Q(\pi))|$ is exactly the generalisation error for an equivariant policy. Taking the supremum, we obtain the equivariant bound (Theorem 5.3) plus $2L_R\xi$. Appendix C.6 provides a detailed proof.

**Corollary 5.7.** *Under the same assumptions as Theorem 5.6, for any* $r \in (0, B)$ *and* $\delta \in (0, 1)$, *the upper bound in Theorem 5.6 for* $\Pi_{approx}(\varepsilon_{eq})$ *is at most the same bound obtained by replacing* $\Pi_{approx}(\varepsilon_{eq})$ *with* $\Pi$. *By Lemma C.8, the return-class covering numbers can be bounded by those of the policy class with radius scaled by* $1/L_R$. *Mathematically, following Theorems 5.1 and 5.5, for any target covering radius* $r > \xi$:

$$\log \mathcal{N}_{\infty,1}\big(\Pi_{approx}(\varepsilon_{eq}), r/L_R\big) \leq \log \mathcal{N}_{\infty,1}\big(\Pi_{eq}, (r-\xi)/L_R\big) \leq \log \mathcal{N}_{\infty,1}\big(\Pi, (r-\xi)/L_R\big), \tag{5}$$

*hence the upper bound in Theorem 5.6 is no larger when evaluated on* $\Pi_{eq}$.

The covering relation incurs a slack of size $\xi$, leading to bounds of the form $N(\Pi_{approx}(\varepsilon_{eq}), r) \leq N(\Pi_{eq}, r - \xi) \leq N(\Pi, r - \xi)$. By contrast, in Corollary 5.4, this slack disappears. Thus, the exact case guarantees a strict reduction in complexity, whereas the approximate case trades a $\xi$-shift in the radius for retaining proximity to the equivariant subspace.

## 6 EXPERIMENTS

We conduct extensive experiments to verify PRISM. The code is at `https://anonymous.4open.science/r/reward_shaping-1CCB`.

## 6.1 EXPERIMENTAL SETTINGS

**Environments.** Four MuJoCo (Todorov et al., 2012) environments are used: mo-hopper-v5, mo-walker2d-v5, mo-halfcheetah-v5, and mo-swimmer-v5. Table 3 in Appendix D displays the environments and their dimensions, highlighting the diversity in space complexity. As a result, a method must be able to find general solutions applicable to various MORL challenges, instead of being just tailored to one specific type of problem. Furthermore, the division of asymmetric and symmetric state and action spaces to model equivariance is detailed in Appendix D.

**Baselines.** PRISM is adaptable to any off-the-shelf MORL algorithm. In this work, CAPQL (Lu et al., 2023) is used as a backbone model, which is a method that trains a single universal network to cover the entire preference space and approximate the Pareto front. We produce (1) **oracle:** instead of artificially setting a reward channel to be sparse, this baseline model can be seen as the gold standard, and (2) **baseline:** instead of utilising the proposed reward shaping model, this method uses CAPQL (Lu et al., 2023) and only observes the sparse rewards.

**Evaluation.** We use hypervolume (HV), Expected Utility Metric (EUM), and one distributional metric, Variance Objective (VO) (Cai et al., 2023), for evaluation. The used hyperparameters, together with a detailed explanation of evaluation metrics, can be found in Appendix E.

## 6.2 EMPIRICAL RESULTS

**Reward Sparsity Sensitivity.** Figure 3 illustrates the sensitivity of MORL agents to varying levels of reward sparsity. Across all environments, we observe a sharp decline in HV when one objective is made extremely sparse, with reductions ranging from 20 to 40% relative to the dense setting. For instance, mo-hopper-v5 exhibits a 35% drop in HV under extreme sparsity, while mo-halfcheetah-v5 and mo-walker2d-v5 show declines of 43% and 21%, respectively. These results confirm that sparse objectives worsen policy quality, as agents tend to neglect long-term sparse signals in favour of denser objectives. For the remainder of the paper, we continue with the most difficult setting where extreme sparsity is imposed on the reward objective along the first dimension.

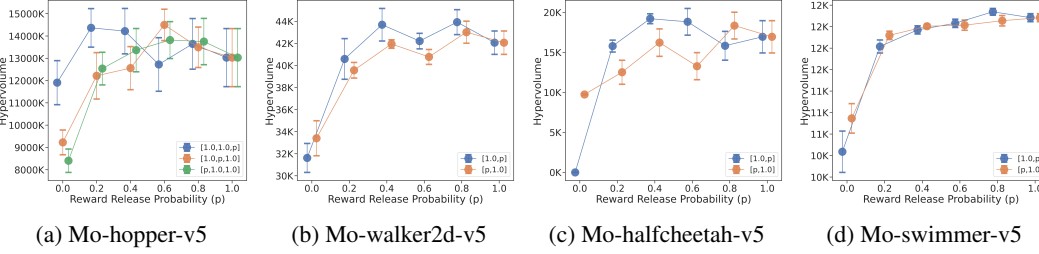

| (a) Mo-hopper-v5 | (b) Mo-walker2d-v5 | (c) Mo-halfcheetah-v5 | (d) Mo-swimmer-v5 |

Figure 3: The obtained hypervolume for various levels of sparsity amongst various dimensions.

**Return Distribution of Policy.** Figure 4 illustrates the impact of mixed sparsity on MORL across the considered environments. Each subplot compares the approximated Pareto fronts obtained when objective one is dense (blue dots) versus when it is made sparse (orange dots), while keeping all other objectives dense. Extreme sparsity is imposed, where the sparse reward is released at the end of an episode. The results demonstrate a consistent pattern across all environments: when objective one becomes sparse, agents systematically fail to discover high-performing solutions along this dimension, instead concentrating their learning efforts on the remaining dense objectives.

The consistent pattern across environments suggests that agents exhibit a systematic bias toward optimising dense reward signals. This overfitting to dense rewards fundamentally distorts the true Pareto front of the problem, leading to a loss of valuable solutions that might represent optimal policies for real-world scenarios where sparse objectives often encode important long-term goals.

**Comparison Experiments.** Table 1 reports the obtained results for HV, EUM, and VO. The results are averaged over 10 trials, with the standard deviations shown in grey.

PRISM consistently outperforms both the oracle and baseline across environments. For mo-hopper-v5, PRISM improves hypervolume by 21.5% over the oracle ($1.58 \times 10^7$ compared to $1.30 \times 10^7$) and 88% over the baseline. Similar gains are observed for mo-walker2d-v5, where PRISM achieves

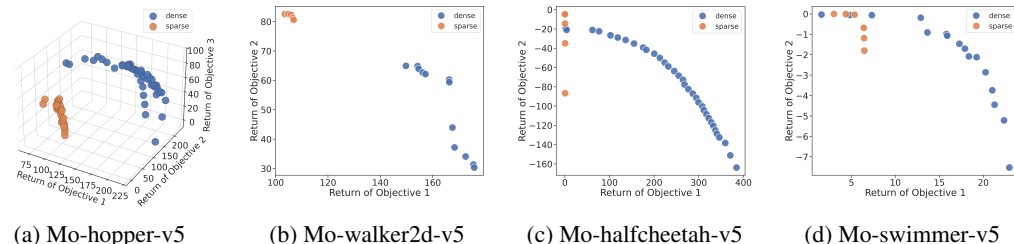

(a) Mo-hopper-v5     (b) Mo-walker2d-v5     (c) Mo-halfcheetah-v5     (d) Mo-swimmer-v5

Figure 4: The approximated Pareto front for dense rewards (blue dots) and sparse rewards (orange dots). Sparsity is imposed on the first reward objective.

Table 1: Experimental results. We report the average hypervolume (HV), Expected Utility Metric (EUM), and Variance Objective (VO) over 10 trials, with the standard error shown in grey. The largest (best) values are in bold font.

| Environment | Metric | Oracle | Baseline | PRISM |
|---|---|---|---|---|
| Mo-hopper-v5 | HV $(\times 10^7)$ | $1.30 \pm 0.13$ | $0.84 \pm 0.05$ | $\mathbf{1.58} \pm 0.05$ |
| | EUM | $129.04 \pm 7.96$ | $97.64 \pm 4.18$ | $\mathbf{147.43} \pm 2.61$ |
| | VO | $59.07 \pm 3.45$ | $43.36 \pm 1.61$ | $\mathbf{66.66} \pm 1.40$ |
| Mo-walker2d-v5 | HV $(\times 10^4)$ | $4.21 \pm 0.11$ | $3.34 \pm 0.16$ | $\mathbf{4.77} \pm 0.07$ |
| | EUM | $107.58 \pm 2.86$ | $82.13 \pm 4.34$ | $\mathbf{120.43} \pm 1.64$ |
| | VO | $53.22 \pm 1.39$ | $39.18 \pm 2.49$ | $\mathbf{59.35} \pm 0.80$ |
| Mo-halfcheetah-v5 | HV $(\times 10^4)$ | $1.70 \pm 0.20$ | $0.97 \pm 0.00$ | $\mathbf{2.25} \pm 0.18$ |
| | EUM | $81.29 \pm 21.85$ | $-1.46 \pm 0.27$ | $\mathbf{89.94} \pm 15.33$ |
| | VO | $36.84 \pm 10.06$ | $-1.01 \pm 0.20$ | $\mathbf{40.72} \pm 7.02$ |
| Mo-swimmer-v5 | HV $(\times 10^4)$ | $\mathbf{1.21} \pm 0.00$ | $1.09 \pm 0.02$ | $\mathbf{1.21} \pm 0.00$ |
| | EUM | $9.41 \pm 0.12$ | $4.10 \pm 0.80$ | $\mathbf{9.44} \pm 0.14$ |
| | VO | $4.22 \pm 0.08$ | $1.58 \pm 0.40$ | $\mathbf{4.24} \pm 0.07$ |

a 13% HV improvement over oracle and 43% over the baseline. Notably, in mo-halfcheetah-v5, PRISM yields a 32% improvement in HV compared to the oracle ($2.25 \times 10^4$ against $1.70 \times 10^4$) and more than doubles the sparse result. These improvements imply that PRISM not only restores solutions lost under sparsity but also expands the range of trade-offs accessible to the agent. Improvements in EUM follow the same trend, with increases of up to 50% compared to the baseline. The concurrent increase in EUM demonstrates that these solutions provide higher expected utility, confirming that PRISM learns policies that are both diverse and practically useful.

On distributional metrics, PRISM delivers more consistent performance than both the oracle and baseline. VO in mo-hopper-v5 increases from 43.36 (baseline) and 59.07 (oracle) to 66.66 under PRISM, and mo-walker2d-v5 shows a 51% gain over the baseline. These gains are crucial because they indicate that PRISM does not simply maximise HV by focusing on extreme solutions, but also produces Pareto fronts that are better balanced, robust, and fair across objectives. Figure 6 in Appendix F, which shows the approximated Pareto fronts, aligns with these results.

We provide two distinct examples to analyse the behaviour of the learned reward signals compared to the oracle for mo-walker2d-v5. Figure 5a illustrates a full 1000-step episode. The shaped reward is highly correlated with the dense reward throughout the entire trajectory. The alignment of peaks and troughs confirms that ReSymNet captures the dynamics of the environment, ensuring accurate credit assignment without temporal drift.

Figure 5b highlights a key theoretical advantage of ReSymNet. In high-performance regions (e.g., steps 250–270), the shaped reward amplifies the signal, exceeding the magnitude of the oracle. By creating steeper gradients for desirable behaviours, the shaped reward can provide more effective guidance than the raw environmental signal, explaining why PRISM is capable of outperforming the oracle.

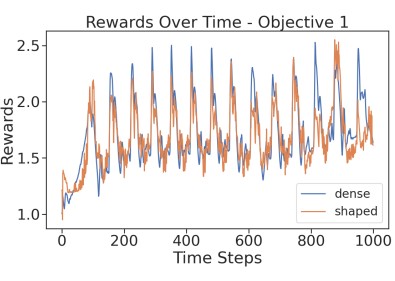 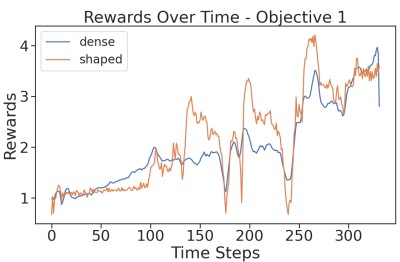

(a) Full-episode stability            (b) Signal optimisation

Figure 5: The dense (blue line) and shaped rewards (orange line) over time for mo-walker2d-v5. Sparsity is imposed on the first reward objective.

**Ablation Study.** We analyse the performance of the following ablation models (w/o is the abbreviation for without), which remove several aspects of the reward shaping model or the equivariance loss: (1) **PRISM:** This is the full proposed framework, (2) **w/o residual:** This ablation model removes the two residual blocks from the reward shaping model, (3) **w/o dense rewards:** We remove the dense rewards as input features to the reward model, (4) **w/o ensemble:** We remove the ensemble of reward shaping models, and only employ one, (5) **w/o refinement:** Rather than updating the reward shaping model with expert trajectories, this approach merely trains the reward shaping model using the random trajectories collected at first, and (6) **w/o loss:** We remove the equivariance loss term and merely use the reward shaping model. We also include two ablation studies that remove ReSymNet from PRISM and replace the reward shaping model as follows: (7) **uniform**: Distributes the episodic sparse reward $R^{\mathrm{sp}}(\tau)$ equally across all $T$ timesteps, and (8) **random**: Samples random weights $\alpha_t \sim \mathcal{U}(-1, 1)$ for each timestep, normalises to sum to one, and scales by the total reward.

The ablation results in Tables 10 and 11 in Appendix G highlight the contribution of individual components. Removing residual connections reduces HV and EUM across all environments (e.g., mo-hopper-v5 EUM falls from 147.43 to 128.40), showing their importance for scaled opportunity value. Excluding dense reward features or ensembles also lowers performance, but only moderately, suggesting that state–action features already contain substantial signal. Interestingly, removing iterative refinement barely reduces performance; in some cases, such as mo-halfcheetah-v5, HV, and EUM remain comparable or even slightly higher than the full model. This implies that shaping rewards from a broad set of random trajectories is already highly effective. Removing the symmetry loss reduces performance across environments, indicating that the loss term successfully reduces the search space. Similar patterns are observed for VO. Considering ReSymNet, uniform achieves moderate performance by providing per-step gradients and leveraging SymReg, while random performs poorly due to noisy, misleading rewards. PRISM consistently outperforms both by learning reward decomposition with ReSymNet and enforcing structural consistency via SymReg, enabling accurate credit assignment in complex multi-objective tasks. The ablation results imply that PRISM's architecture provides multiple overlapping mechanisms for stability, but the symmetry loss and residual structure are the main drivers of consistent performance.

## 7 CONCLUSION

This work proposes Parallel Reward Integration with reflectional Symmetry for Multi-objective reinforcement learning (PRISM), a framework designed to tackle sample inefficiency in heterogeneous multi-objective reinforcement learning, particularly in environments with sparse rewards. Our approach is centred around two key contributions: (1) ReSymNet, a theory-inspired reward model that leverages residual blocks to align reward channels by learning a refined 'scaled opportunity value', and (2) SymReg, a novel regulariser that enforces reflectional symmetry as an inductive bias in the policy's action space. We prove that PRISM restricts policy search to a reflection-equivariant subspace, a projection of the original policy space with provably reduced hypothesis complexity; in this way, the generalisability is rigorously improved. Extensive experiments on MuJoCo benchmarks show that PRISM consistently outperforms even a strong oracle with full reward access in terms of a wide range of metrics, including HV, EUM, and VO.

## ETHICS STATEMENT

We declare no potential conflict of interest nor sponsorship. We are not aware of any issues related to legal compliance, research integrity, or other ethical considerations.

## REPRODUCIBILITY STATEMENT

We have taken several steps to ensure reproducibility. All assumptions underlying our theoretical results are explicitly stated in Section 5. For the empirical results, we use only publicly available environments, described in Section 6, with training details, hyperparameters, and evaluation metrics reported in Appendix E. To further support reproducibility, we released an anonymous code repository containing implementation details at `https://anonymous.4open.science/r/reward_shaping-1CCB`, which will be publicly released.

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

## A NOTATION

In this appendix, we provide an overview of the notation used in Table 2.

Table 2: Notation.

| Symbol | Meaning |
|---|---|
| $\mathcal{S}$ | State space |
| $\mathcal{A}$ | Action space |
| $P(s'\|s,a)$ | Transition probability |
| $\boldsymbol{r}(s,a) \in \mathbb{R}^L$ | Vector-valued reward with $L$ objectives |
| $\gamma \in [0,1)$ | Discount factor |
| $\pi : \mathcal{S} \to \mathcal{A}$ | Policy mapping |
| $\boldsymbol{J}(\pi) = \mathbb{E}_\pi\big[\sum_{t=0}^\infty \gamma^t \boldsymbol{r}_t\big]$ | Expected cumulative vector return |
| $\mathcal{D}$ | Behaviour distribution to sample episodes from |
| $DC = \{d_1, \ldots, d_D\}$ | Dense reward channels |
| $r_t^{d_i}$ | Reward from dense channel $d_i$ at timestep $t$ |
| $r_t^{sp}$ | Sparse reward at timestep $t$ |
| $\tau = \{(s_1, a_1), \ldots, (s_T, a_T)\}$ | Trajectory |
| $R^{sp}(\tau)$ | Cumulative sparse reward in episode $\tau$ |
| $p_{\text{rel}}$ | Probability of releasing sparse reward |
| $h_t = [s_t, a_t, \boldsymbol{r}_t^{\text{dense}}]$ | Input feature vector for ReSymNet |
| $\mathcal{R}_{\text{pred}}$ | ReSymNet |
| $r_t^{sh}$ | Shaped reward at timestep $t$ |
| $L_g, K_g$ | Reflection operators on states and actions |
| $\Delta_\pi(s) = \pi(L_g(s)) - K_g(\pi(s))$ | Equivariance mismatch |
| $\mathcal{L}_{eq}$ | Equivariance regularisation loss |
| $\Pi$ | Hypothesis space of policies |
| $\Pi_{eq} = \{\pi : \pi(L_g(s)) = K_g(\pi(s))\}$ | Reflection-equivariant subspace |
| $\Pi_{\text{approx}}(\varepsilon_{eq})$ | Approximate equivariant policies with tolerance $\varepsilon_{eq}$ |

# B  ADDITIONAL DETAILS AND THEORY OF RESYMNET

We give additional details of ReSymNet as well as the theoretical motivation behind its architecture in this appendix.

## B.1  THEORETICAL MOTIVATION VIA SCALED OPPORTUNITY VALUE

The use of residual connections in $\mathcal{R}_{\text{pred}}$ is motivated by the theory of scaled opportunity value (Laud, 2004).

**Definition 6** (Opportunity value). *Let $M$ be an MDP with native reward function $R$. The opportunity value of a transition $(s, a, s')$ is defined as the difference in the optimal value of successor and current states: $\text{OPV}(s, a, s') = \gamma V^M(s') - V^M(s)$, where $V^M$ is the optimal state-value function under MDP $M$.*

**Definition 7** (Scaled opportunity value). *For a scale parameter $k > 0$, the scaled opportunity value shaping function augments the native reward with a scaled opportunity correction: $\text{OPV}_k(s, a, s') = F_k(s, a, s') = k(\gamma V^M(s') - V^M(s)) + (k - 1)R(s, a)$.*

**Lemma B.1.** *Let $M$ be an MDP with reward function $R$ and optimal policy $\pi^\star$. With $k$ sufficiently large, the MDP with shaped reward $F_k$ satisfies: (1) policy invariance, $\pi^\star$ remains optimal under $F_k$; (2) horizon reduction, the effective reward horizon is reduced to 1; and (3) improved local approximation, the additive term increases the separability of local utilities, reducing approximation error in value estimation.*

Residual blocks mirror the additive structure of scaled opportunity value: each block refines its input prediction via: $\mathcal{R}_{\text{pred}}^{(i)}(\boldsymbol{h}_t; \psi) = \mathcal{R}_{\text{pred}}^{(i-1)}(\boldsymbol{h}_t; \psi) + \Delta_i(\boldsymbol{h}_t; \psi)$, where $\Delta_i$ is a learned correction. A single block can be viewed as approximating a scaled opportunity-value transformation of its input, while stacking multiple blocks implements iterative refinement: each stage reduces the residual error left by the previous one. This residual formulation both stabilises training and aligns with the principle of scaled opportunity value, gradually shaping per-step predictions into horizon-1 signals that remain consistent with the sparse episodic return $R^{sp}(\tau)$.

## B.2  GENERALISABILITY OF RESYMNET

We extend the theoretical justification of ReSymNet from optimisation to generalisation. Following the stem–vine decomposition of He et al. (2020), we prove that residual connections do not increase hypothesis complexity, and derive a high-probability bound.

**Notation and Assumptions.** ReSymNet maps feature vectors $\mathbf{h}_t \in \mathbb{R}^{d_0}$ to sparse reward predictions $r_t^{\text{sp}} \in \mathbb{R}$ through a residual network. We decompose the network into:

- A stem: the main feedforward pathway consisting of $K$ layers, each with a weight matrix $\mathbf{A}_i \in \mathbb{R}^{d_{i-1} \times d_i}$ and nonlinearity $\sigma_i : \mathbb{R}^{d_i} \to \mathbb{R}^{d_i}$ for $i = 1, \ldots, K$.

- A collection of vines: residual connections (skip connections) indexed by triples $(s, t, i)$ where $s$ is the source vertex (where the connection starts), $t$ is the target vertex (where it reconnects), and $i$ distinguishes multiple vines between the same pair of vertices. We denote the set of all vine indices as $\mathcal{I}_V$.

We denote vertices in the network as $N(t)$, where $t$ indexes the position in the computational graph. Each vine $\mathcal{V}(s, t, i)$ is itself a small feedforward network with weight matrices $\mathbf{A}_1^{s,t,i}, \ldots, \mathbf{A}_{K_{s,t,i}}^{s,t,i}$ and nonlinearities $\sigma_1^{s,t,i}, \ldots, \sigma_{K_{s,t,i}}^{s,t,i}$, where $K_{s,t,i}$ is the number of layers in that vine. The output at vertex $N(t)$ is:

$$F_t(\mathbf{X}) = F_t^S(\mathbf{X}) + \sum_{(s,t,i) \in \mathcal{I}_V} F_{s,t,i}^V(\mathbf{X}),$$

where $F_t^S(\mathbf{X})$ is the stem's output at vertex $t$ and the sum runs over all vines that reconnect at vertex $t$.

**Assumption 6** (Bounded parameters). *Each stem weight matrix satisfies $\|\mathbf{A}_i\|_\sigma \leq s_i$ for $i = 1, \ldots, K$, where $\|\cdot\|_\sigma$ denotes the spectral norm. Each vine weight matrix satisfies $\|\mathbf{A}_j^{s,t,i}\|_\sigma \leq$*

$s_j^{s,t,i}$. *All nonlinearities are $\rho_i$-Lipschitz continuous: for any $\mathbf{x}_1, \mathbf{x}_2$ in the domain,*

$$\|\sigma_i(\mathbf{x}_1) - \sigma_i(\mathbf{x}_2)\|_2 \leq \rho_i \|\mathbf{x}_1 - \mathbf{x}_2\|_2.$$

*Input features satisfy $\|\mathbf{h}_t\|_2 \leq B_h$, network per-step outputs satisfy $|R_{pred}(\mathbf{h}_t; \psi)| \leq B_{pred}$ for all $t$, and sparse rewards satisfy $|R^{sp}(\tau)| \leq B_r$ for all trajectories $\tau$. Trajectories have length bounded by $T_{\max}$.*

**Lemma B.2.** *Let $\mathbf{X} \in \mathbb{R}^{n \times d}$ be a data matrix with $n$ samples and $d$ features, satisfying $\|\mathbf{X}\|_2 \leq B$. Consider the hypothesis space formed by all linear transformations with bounded spectral norm:*

$$\mathcal{H}_A = \{\mathbf{X}\mathbf{A} : \mathbf{A} \in \mathbb{R}^{d \times m}, \ \|\mathbf{A}\|_\sigma \leq s\}.$$

*Then the $\varepsilon$-covering number satisfies:*

$$\log \mathcal{N}_{\infty,2}(\mathcal{H}_A, \varepsilon) \leq \left\lceil \frac{s^2 B^2 m^2}{\varepsilon^2} \right\rceil \log(2dm),$$

*where $m$ is the output dimension.*

This lemma (Bartlett et al., 2017) shows that the complexity of a single linear layer scales with the square of its spectral norm and input norm.

**Lemma B.3.** *For an $K$-layer feedforward network with hypothesis space $\mathcal{H}_{\mathrm{ff}}$, the covering number satisfies:*

$$\mathcal{N}_{\infty,2}(\mathcal{H}_{\mathrm{ff}}, \varepsilon) \leq \prod_{i=1}^{K} \sup_{\mathbf{A}_1, \ldots, \mathbf{A}_{i-1}} \mathcal{N}_i,$$

*where $\mathcal{N}_i$ is the covering number of layer $i$ (viewed as a function of its input) when the preceding layers $\mathbf{A}_1, \ldots, \mathbf{A}_{i-1}$ are held fixed. The supremum is taken over all choices of the preceding weight matrices within their respective spectral norm bounds.*

This result shows that the covering number of a deep network is the product of the covering numbers of its individual layers. For residual networks, where outputs are sums of stem and vine contributions, we require:

**Lemma B.4.** *Let $\mathcal{F}$ and $\mathcal{G}$ be two function classes. If $\mathcal{W}_F$ is an $\varepsilon_F$-cover of $\mathcal{F}$ (meaning every $f \in \mathcal{F}$ is within distance $\varepsilon_F$ of some element in $\mathcal{W}_F$), and $\mathcal{W}_G$ is an $\varepsilon_G$-cover of $\mathcal{G}$, then the set*

$$\mathcal{W}_F + \mathcal{W}_G = \{f + g : f \in \mathcal{W}_F, \ g \in \mathcal{W}_G\}$$

*is an $(\varepsilon_F + \varepsilon_G)$-cover of the sum class $\mathcal{F} + \mathcal{G} = \{f + g : f \in \mathcal{F}, g \in \mathcal{G}\}$, and*

$$\mathcal{N}_{\infty,2}(\mathcal{F} + \mathcal{G}, \varepsilon_F + \varepsilon_G) \leq \mathcal{N}_{\infty,2}(\mathcal{F}, \varepsilon_F)\mathcal{N}_{\infty,2}(\mathcal{G}, \varepsilon_G).$$

*Proof.* For any $f + g \in \mathcal{F} + \mathcal{G}$, there exist $w_f \in \mathcal{W}_F$ and $w_g \in \mathcal{W}_G$ such that $\|f - w_f\|_2 \leq \varepsilon_F$ and $\|g - w_g\|_2 \leq \varepsilon_G$. By the triangle inequality:

$$\|(f + g) - (w_f + w_g)\|_2 \leq \|f - w_f\|_2 + \|g - w_g\|_2 \leq \varepsilon_F + \varepsilon_G.$$

The covering number bound follows since there are at most $|\mathcal{W}_F| \cdot |\mathcal{W}_G|$ distinct pairs $(w_f, w_g)$. $\square$

**Theorem B.5.** *Under Assumption 6, let $\{\varepsilon_j\}_{j=1}^{K}$ be tolerances for each stem layer and $\{\varepsilon_{s,t,i}\}_{(s,t,i) \in \mathcal{I}_V}$ be tolerances for each vine, satisfying*

$$\sum_{j=1}^{K} \varepsilon_j + \sum_{(s,t,i) \in \mathcal{I}_V} \varepsilon_{s,t,i} \leq \varepsilon.$$

*Then the covering number of ReSymNet's hypothesis space $\mathcal{H}_{\mathrm{res}}$ satisfies:*

$$\mathcal{N}_{\infty,2}(\mathcal{H}_{\mathrm{res}}, \varepsilon) \leq \prod_{j=1}^{K} \mathcal{N}_{\infty,2}(\mathcal{H}_j, \varepsilon_j) \prod_{(s,t,i) \in \mathcal{I}_V} \mathcal{N}_{\infty,2}(\mathcal{H}_{s,t,i}^V, \varepsilon_{s,t,i}),$$

*where $\mathcal{H}_j$ is the hypothesis space of stem layer $j$ and $\mathcal{H}_{s,t,i}^V$ is the hypothesis space of vine $\mathcal{V}(s,t,i)$. Applying Lemma B.2 to each weight matrix, this yields:*

$$\log \mathcal{N}_{\infty,2}(\mathcal{H}_{\mathrm{res}}, \varepsilon) \leq \frac{\mathcal{R}}{\varepsilon^2},$$

*where the complexity measure $\mathcal{R}$ is:*

$$\mathcal{R} = \sum_{i=1}^{K} \frac{s_i^2 \|F_{i-1}(\mathbf{X})\|_2^2}{\varepsilon_i^2} \log(2d_i^2) + \sum_{(s,t,i) \in \mathcal{I}_V} \frac{(s^{s,t,i})^2 \|F_s(\mathbf{X})\|_2^2}{\varepsilon_{s,t,i}^2} \log(2d_{s,t,i}^2).$$

*Here, $F_{i-1}(\mathbf{X})$ denotes the output of the network after layer $i-1$ (the input to layer $i$), and $d_i$ is the dimension at layer $i$.*

*Proof.* We proceed by analysing how residual connections compose with the stem. Consider vertex $N(t)$ where one or more vines reconnect. The output is:

$$F_t(\mathbf{X}) = F_t^S(\mathbf{X}) + \sum_{(s,t,i) \in \mathcal{I}_V} F_{s,t,i}^V(\mathbf{X}).$$

Let $\mathcal{W}_t$ be an $\varepsilon_t$-cover of $\mathcal{H}_t$ (all possible stem outputs at vertex $t$). For each vine $\mathcal{V}(s,t,i)$ that reconnects at $t$, let $\mathcal{W}_{s,t,i}^V$ be an $\varepsilon_{s,t,i}$-cover of $\mathcal{H}_{s,t,i}^V$ (all possible outputs of that vine). By repeated application of Lemma B.4, the set:

$$\mathcal{W}_t' = \left\{ W_S + \sum_{(s,t,i) \in \mathcal{I}_V} W_{s,t,i}^V : W_S \in \mathcal{W}_t, W_{s,t,i}^V \in \mathcal{W}_{s,t,i}^V \right\}$$

is an $\left( \varepsilon_t + \sum_{(s,t,i) \in \mathcal{I}_V} \varepsilon_{s,t,i} \right)$-cover of $\mathcal{H}_t'$ (the combined outputs at vertex $t$), with covering number:

$$\mathcal{N}_{\infty,2}(\mathcal{H}_t', \varepsilon_t') \leq \mathcal{N}_{\infty,2}(\mathcal{H}_t, \varepsilon_t) \cdot \prod_{(s,t,i) \in \mathcal{I}_V} \mathcal{N}_{\infty,2}(\mathcal{H}_{s,t,i}^V, \varepsilon_{s,t,i}),$$

where $\varepsilon_t' = \varepsilon_t + \sum_{(s,t,i) \in \mathcal{I}_V} \varepsilon_{s,t,i}$.

Each vine $\mathcal{V}(s,t,i)$ is itself a chain-like feedforward network, so Lemma B.3 applies to bound $\mathcal{N}_{\infty,2}(\mathcal{H}_{s,t,i}^V, \varepsilon_{s,t,i})$. For identity vines (containing no trainable parameters), we have $\mathcal{N}_{s,t,i}^V = 1$ since there is only one function in the class.

Propagating this argument through all $K$ stem layers yields:

$$\mathcal{N}_{\infty,2}(\mathcal{H}_{\mathrm{res}}, \varepsilon) \leq \prod_{j=1}^{K} \mathcal{N}_{\infty,2}(\mathcal{H}_j, \varepsilon_j) \prod_{(s,t,i) \in \mathcal{I}_V} \mathcal{N}_{\infty,2}(\mathcal{H}_{s,t,i}^V, \varepsilon_{s,t,i}).$$

The bound on $\mathcal{R}$ follows by applying Lemma B.2 to each weight matrix. For the stem, layer $i$ contributes:

$$\log \mathcal{N}_i \leq \frac{s_i^2 \|F_{i-1}(\mathbf{X})\|_2^2 d_i^2}{\varepsilon_i^2} \log(2d_{i-1}d_i) \approx \frac{s_i^2 \|F_{i-1}(\mathbf{X})\|_2^2}{\varepsilon_i^2} \log(2d_i^2),$$

where we simplify by assuming similar dimensions. Summing over all stem layers and all vine layers gives $\mathcal{R}$. $\square$

**Corollary B.6.** *Let $\mathcal{H}_{\mathrm{ff}}$ be the hypothesis space of feedforward networks with the same total number of weight matrices $K_{total} = K + \sum_{(s,t,i) \in \mathcal{I}_V} K_{s,t,i}$ as ReSymNet. Then for any $\varepsilon > 0$,*

$$\mathcal{N}_{\infty,2}(\mathcal{H}_{\mathrm{res}}, \varepsilon) \leq \mathcal{N}_{\infty,2}(\mathcal{H}_{\mathrm{ff}}, \varepsilon).$$

*Proof.* Both covering numbers have the product form $\prod_{k=1}^{K_{\mathrm{total}}} \mathcal{N}_k$, where each factor $\mathcal{N}_k$ corresponds to a single weight matrix. By Lemma B.2, each $\mathcal{N}_k$ depends only on the spectral norm $s_k$ of that weight matrix and the norm of its input $\|F_{k-1}(\mathbf{X})\|_2$, regardless of whether the matrix appears in the stem or a vine. Therefore, when the total number of weight matrices and their norms are held fixed, the covering numbers are bounded identically. $\square$

### B.3 ALGORITHM CHART

---

**Algorithm 1:** ReSymNet with any MORL algorithm

---

**Input:** Release probability $p_{\text{rel}}$, number of initial episodes $N$, number of expert episodes $E$, dense channels $\mathcal{DC}$, any off-the-shelf MORL algorithm, number of timesteps per cycle $M$, number of ensembles $K$, number of iterative refinements $IR$, validation split, patience

**Output:** Trained reward ensemble $\mathcal{E} = \{\mathcal{R}_{\text{pred},\psi_1}, \ldots, \mathcal{R}_{\text{pred},\psi_K}\}$, trained MORL policy

/* Collecting random experiences */

**for** $i \leftarrow 1$ **to** $N$ **do**

    Execute a random policy to collect trajectory $\tau = \{(s_0, a_0), \ldots, (s_T, a_T)\}$ until episode ends

    Set $l = 0$

    **foreach** $t \in T$ **do**

        With probability $p_{\text{rel}}$, release cumulative sparse reward $R_t^{sp} = \sum_{s=l}^{t} r_s^{sp}$ at timestep $t$

        Set $l = t$ if sparse reward is released

    Segment $\tau$ into sub-trajectories $\{\tau_j\}$ based on released rewards

    **foreach** *sub-trajectory* $\tau_j$ **do**

        **foreach** $(s_t, a_t) \in \tau_j$ **do**

            Compute features: $\boldsymbol{h}_t = [s_t, a_t, \boldsymbol{r}_t^{\text{dense}}]$

        Add datapoint $\left(\{\boldsymbol{h}_t\}_{t \in \tau_j}, R^{sp}(\tau_j)\right)$ to dataset $\mathcal{D}$

/* Ensemble training */

**for** $k \leftarrow 1$ **to** $K$ **do**

    Split $\mathcal{D}$ into $\mathcal{D}_{\text{train}}$ and $\mathcal{D}_{val}$ using the validation split

    Train reward model $\mathcal{R}_{\text{pred},\psi_k}$ following Equation 1 using early stopping on the validation loss:

$$\mathcal{L}(\psi_k) = \sum_{\tau \in \mathcal{D}_{\text{train}}} \left( \sum_{t \in \tau} \mathcal{R}_{\text{pred}}(\boldsymbol{h}_t; \psi_k) - R^{sp}(\tau) \right)^2$$

/* RL training with iterative refinement */

$timestep = 1$

**for** $cycle \leftarrow 1$ **to** $IR$ **do**

    **for** $t \leftarrow timestep$ **to** $M + timestep$ **do**

        Observe $s_t$ and $a_t$ following the current policy and compute features $\boldsymbol{h}_t$

        $r_t^{(k)} \leftarrow \mathcal{R}_{\text{pred}}(\boldsymbol{h}_t; \psi_k)$ for $k = 1, \ldots, K$

        $r_t^{\text{sh}} \leftarrow \frac{1}{K} \sum_{k=1}^{K} r_t^{(k)}$

        Use $r_t^{\text{sh}}$ with the dense rewards as the reward at timestep $t$ and update RL algorithm

    /* Iterative refinement */

    Collect $E$ expert trajectories to obtain $\mathcal{D}_{\text{new}}$ using the new policy

    **foreach** $\mathcal{R}_{pred,\psi_k} \in \mathcal{E}$ **do**

        Update $\mathcal{R}_{\text{pred},\psi_k}$ using new data $\mathcal{D}_{\text{new}}$

    $timestep = t$

---

## C PROOFS

This appendix collects all proofs omitted from the main text.

### C.1 LEMMAS

This section introduces the general lemmas used to obtain an upper bound on the generalisation gap.

**Dudley Entropy Integral.** The Rademacher complexity can be bounded through the metric entropy of the function class using Dudley's entropy integral (Dudley, 1967; Bartlett & Mendelson, 2002).

**Lemma C.1** (Dudley Entropy Integral). *For any coarse-scale parameter $r \in (0, B)$, the empirical Rademacher complexity satisfies:*

$$\hat{\mathfrak{R}}_N(\mathcal{F}) \leq C \left( \int_r^B \sqrt{\frac{\log \mathcal{N}_{\infty,1}(\mathcal{F}, r)}{N}} d\varepsilon \right) + \frac{4r}{\sqrt{N}},$$

*where $C > 0$ is an absolute constant, and $\mathcal{N}_{\infty,1}(\mathcal{F}, r)$ is the covering number of $\mathcal{F}$ in $\ell_\infty$ at scale $r$ with respect to $N$ samples*

This inequality connects the probabilistic complexity (Rademacher complexity) to the geometric complexity of the function class and covering numbers.

**McDiarmid's Concentration Inequality.** To convert expectation bounds into high-probability statements, we employ McDiarmid's bounded difference inequality (McDiarmid et al., 1989).

**Lemma C.2** (McDiarmid's Concentration Inequality). *If each trajectory's replacement can change any empirical average by at most $B/N$, then for any $t > 0$:*

$$\Pr\left(\left|\sup_{f \in \mathcal{F}} \frac{1}{N} \sum_{i=1}^{N}(f(\tau_i) - \mathbb{E}[f]) - \mathbb{E}\left[\sup_{f \in \mathcal{F}} \frac{1}{N} \sum_{i=1}^{N}(f(\tau_i) - \mathbb{E}[f])\right]\right| \geq t\right) \leq 2\exp\left(-\frac{2Nt^2}{B^2}\right).$$

This concentration result allows us to bound the deviation between the random supremum and its expectation, completing the pipeline from covering numbers to high-probability uniform generalisation gaps.

## C.2 GENERALISATION OF SCALARISED RETURNS

This section shows that generalisation for an arbitrary scalar return implies guarantees for the scalarised components of the Pareto front.

**Corollary C.3.** *Let $\Pi$ be a policy class equipped with a metric $d(\cdot, \cdot)$, and let $\boldsymbol{R}(\pi; \tau) \in \mathbb{R}^L$ denote the vector-valued return of policy $\pi$ on trajectory $\tau$. Following Assumption 1:*

$$\sup_{\tau} \|\boldsymbol{R}(\pi; \tau) - \boldsymbol{R}(\tilde{\pi}; \tau)\|_\infty \leq L_R\, d(\pi, \tilde{\pi}) \qquad \text{for all } \pi, \tilde{\pi} \in \Pi.$$

*For any weight vector $\omega \in \mathbb{R}^L$ define the scalarised return $R^\omega(\pi; \tau) = \omega^\top \boldsymbol{R}(\pi; \tau)$ and let $\mathcal{R}_\Pi^\omega$ be the class of scalarised returns induced by $\Pi$. Then for any $\varepsilon > 0$,*

$$\mathcal{N}_{\infty,1}(\mathcal{R}_\Pi^\omega, \varepsilon) \leq \mathcal{N}_{\infty,1}\big(\Pi,\ \varepsilon/L_R^\omega\big), \qquad \text{where } L_R^\omega := \|\omega\|_1 L_R.$$

*In particular, when $\|\omega\|_1 = 1$ we have $L_R^\omega = L_R$ and the scalarised return class has covering numbers no larger than those of the policy class. Consequently, any complexity reduction obtained by projecting $\Pi$ to an equivariant subspace (e.g. $\Pi_{eq}$) is inherited by the scalarised objective class $\mathcal{R}_\Pi^\omega$.*

*Proof.* Fix $\omega \in \mathbb{R}^L$ and let $\pi, \tilde{\pi} \in \Pi$. For any trajectory $\tau$,

$$\left|R^\omega(\pi; \tau) - R^\omega(\tilde{\pi}; \tau)\right| = \left|\omega^\top\big(\boldsymbol{R}(\pi; \tau) - \boldsymbol{R}(\tilde{\pi}; \tau)\big)\right| \leq \sum_{j=1}^{L} |\omega_j|\, \left|R_j(\pi; \tau) - R_j(\tilde{\pi}; \tau)\right|.$$

Using $\max_j |R_j(\pi; \tau) - R_j(\tilde{\pi}; \tau)| = \|\boldsymbol{R}(\pi; \tau) - \boldsymbol{R}(\tilde{\pi}; \tau)\|_\infty$, we obtain

$$\left|R^\omega(\pi; \tau) - R^\omega(\tilde{\pi}; \tau)\right| \leq \|\omega\|_1 \|\boldsymbol{R}(\pi; \tau) - \boldsymbol{R}(\tilde{\pi}; \tau)\|_\infty.$$

Taking the supremum over trajectories and applying the vector Lipschitz assumption yields

$$\sup_{\tau} \left|R^\omega(\pi; \tau) - R^\omega(\tilde{\pi}; \tau)\right| \leq \|\omega\|_1 L_R\, d(\pi, \tilde{\pi}) = L_R^\omega\, d(\pi, \tilde{\pi}).$$

Thus the scalarised return map $\pi \mapsto R^\omega(\pi; \cdot)$ is Lipschitz with constant $L_R^\omega = \|\omega\|_1 L_R$. Following Lemma C.8, for any $\varepsilon > 0$,

$$\mathcal{N}_{\infty,1}(\mathcal{R}_\Pi^\omega, \varepsilon) \leq \mathcal{N}_{\infty,1}\big(\Pi,\ \varepsilon/L_R^\omega\big).$$

This proves the displayed inequality. The special case $\|\omega\|_1 = 1$ follows immediately. Finally, since the inequality holds for any policy class $\Pi$, replacing $\Pi$ by the equivariant subspace $\Pi_{eq}$ shows that any complexity reduction ($\mathcal{N}(\Pi_{eq}, \cdot)$ smaller than is directly inherited by the scalarised return class. $\square$

### C.3 PROJECTION TO REFLECTION-EQUIVARIANT SUBSPACE

Let the full hypothesis space of policies be $\Pi = \{\pi_\phi : \phi \in \Phi\}$, where $\phi$ represents the neural network parameters and $\Phi$ represents the parameter space. The reflection group $G = \mathbb{Z}_2 = \{e, g\}$ acts on the state and action spaces via operators $L_g$ and $K_g$, respectively.

We can map any policy to its equivariant counterpart using an orbit averaging operator $\mathcal{Q} : \Pi \to \Pi$, defined as:

$$
\begin{aligned}
\mathcal{Q}(\pi_\phi)(s) &= \frac{1}{|G|} \sum_{h \in G} \rho(h) \pi_\phi(h^{-1} \cdot s) \\
&= \frac{1}{|G|} \sum_{h \in G} K_h\big(\pi_\phi(L_h(s))\big) \\
&= \tfrac{1}{2}\left(\pi_\phi(s) + K_g(\pi_\phi(L_g(s)))\right).
\end{aligned}
\tag{6}
$$

Here, $\rho(h)$ is the abstract representation in the action space, and $h^{-1} \cdot s$ is the abstract action in the state space. In the second line we replace $\rho(h)$ with the action transformation $K_h$, and $h^{-1} \cdot s$ with the state transformation $L_h(s)$. For the reflection group $G = \mathbb{Z}_2 = \{e, g\}$, since $g = g^{-1}$ we may drop the inverse without ambiguity. This operator averages a policy's output with its reflected-transformed equivalent. The regulariser, $\mathcal{L}_{\text{eq}} = \mathbb{E}_s[\|\pi_\phi(L_g(s)) - K_g(\pi_\phi(s))\|_1^2]$, encourages policies to become fixed points of this operator, thereby learning policies within the subspace of equivariant functions, denoted $\Pi_{\text{eq}}$.

The operator $\mathcal{Q}$ and the subspace $\Pi_{\text{eq}}$ have several crucial properties, which we state in the following lemmas.

**Lemma C.4.** *For any $\pi \in \Pi$, the function $\mathcal{Q}(\pi)$ is reflectional equivariant:*

$$
\mathcal{Q}(\pi)(L_g(s)) = K_g(\mathcal{Q}(\pi)(s)), \quad \forall s \in \mathcal{S}.
$$

*Proof.* By direct calculation:

$$
\begin{aligned}
\mathcal{Q}(\pi)(L_g(s)) &= \tfrac{1}{2}\big(\pi(L_g(s)) + K_g(\pi(L_g(L_g(s))))\big) \\
&= \tfrac{1}{2}\big(\pi(L_g(s)) + K_g(\pi(s))\big), \\
K_g\mathcal{Q}(\pi)(s) &= \tfrac{1}{2}\big(K_g(\pi(s)) + K_g K_g(\pi(L_g(s)))\big) \\
&= \tfrac{1}{2}\big(K_g(\pi(s)) + \pi(L_g(s))\big),
\end{aligned}
$$

since $K_g$ and $L_g$ are involutions. Thus, the two expressions coincide. Therefore $\mathcal{Q}(\pi)$ is equivariant. $\square$

**Lemma C.5.** *The operator $\mathcal{Q}$ is a projection, meaning it is idempotent: $\mathcal{Q}(\mathcal{Q}(\pi)) = \mathcal{Q}(\pi)$ for any $\pi \in \Pi$.*

*Proof.* We apply the operator to its own output:

$$
\mathcal{Q}(\mathcal{Q}(\pi))(s) = \frac{1}{2}\left(\mathcal{Q}(\pi)(s) + K_g(\mathcal{Q}(\pi)(L_g(s)))\right).
$$

First, evaluating the second term, $\mathcal{Q}(\pi)(L_g(s))$:

$$
\begin{aligned}
\mathcal{Q}(\pi)(L_g(s)) &= \frac{1}{2}\left(\pi(L_g(s)) + K_g(\pi(L_g(L_g(s))))\right) \\
&= \frac{1}{2}\left(\pi(L_g(s)) + K_g(\pi(s))\right).
\end{aligned}
$$

Substituting this back:

$$
\begin{aligned}
\mathcal{Q}(\mathcal{Q}(\pi))(s) &= \frac{1}{2}\left(\mathcal{Q}(\pi)(s) + K_g\left[\tfrac{1}{2}(\pi(L_g(s)) + K_g(\pi(s)))\right]\right) \\
&= \frac{1}{2}\mathcal{Q}(\pi)(s) + \frac{1}{4}\left(K_g(\pi(L_g(s))) + K_g(K_g(\pi(s)))\right) \\
&= \frac{1}{2}\mathcal{Q}(\pi)(s) + \frac{1}{4}\left(K_g(\pi(L_g(s))) + \pi(s)\right) \\
&= \frac{1}{2}\mathcal{Q}(\pi)(s) + \frac{1}{2}\left(\tfrac{1}{2}(\pi(s) + K_g(\pi(L_g(s))))\right) \\
&= \frac{1}{2}\mathcal{Q}(\pi)(s) + \frac{1}{2}\mathcal{Q}(\pi)(s) \\
&= \mathcal{Q}(\pi)(s).
\end{aligned}
$$

Thus $\mathcal{Q}$ is idempotent. $\qquad\square$

**Lemma C.6.** *The image of the operator $\mathcal{Q}$ coincides with the set of equivariant policies:* $\mathrm{Im}(\mathcal{Q}) = \{\mathcal{Q}(\pi) : \pi \in \Pi\} = \Pi_{eq}.$

*Proof.* We establish set equality by showing inclusion in both directions.

First inclusion ($\mathrm{Im}(\mathcal{Q}) \subseteq \Pi_{\text{eq}}$): By Lemma C.4, for any $\pi \in \Pi$, the output $\mathcal{Q}(\pi)$ is equivariant. Therefore, every element in the image of $\mathcal{Q}$ belongs to $\Pi_{\text{eq}}$.

Second inclusion ($\Pi_{\text{eq}} \subseteq \mathrm{Im}(\mathcal{Q})$): Let $\pi_{\text{eq}}$ be any equivariant policy, so $\pi_{\text{eq}} \in \Pi_{\text{eq}}$. We need to show that $\pi_{\text{eq}}$ can be expressed as $\mathcal{Q}(\pi)$ for some $\pi \in \Pi$.

Since $\pi_{\text{eq}}$ is equivariant, it satisfies $\pi_{\text{eq}}(L_g(s)) = K_g(\pi_{\text{eq}}(s))$ for all $s$. Therefore:

$$
\begin{aligned}
\mathcal{Q}(\pi_{\text{eq}})(s) &= \frac{1}{2}\left(\pi_{\text{eq}}(s) + K_g(\pi_{\text{eq}}(L_g(s)))\right) \\
&= \frac{1}{2}\left(\pi_{\text{eq}}(s) + K_g(K_g(\pi_{\text{eq}}(s)))\right) \quad \text{(by equivariance)} \\
&= \frac{1}{2}\left(\pi_{\text{eq}}(s) + \pi_{\text{eq}}(s)\right) \quad \text{(since } K_g \text{ is an involution)} \\
&= \pi_{\text{eq}}(s).
\end{aligned}
$$

Therefore, $\pi_{\text{eq}} = \mathcal{Q}(\pi_{\text{eq}}) \in \mathrm{Im}(\mathcal{Q})$. This shows that equivariant policies are fixed points of $\mathcal{Q}$, which is consistent with Lemma C.5. Since every equivariant policy is its own image under $\mathcal{Q}$, we have $\Pi_{\text{eq}} \subseteq \mathrm{Im}(\mathcal{Q})$. Combining both inclusions yields $\mathrm{Im}(\mathcal{Q}) = \Pi_{\text{eq}}$. Therefore $\mathcal{Q}$ is surjective onto $\Pi_{\text{eq}}$. $\qquad\square$

## C.4 REDUCED HYPOTHESIS COMPLEXITY OF REFLECTION-EQUIVARIANT SUBSPACE

To prove that the subspace $\Pi_{\text{eq}}$ is less complex, we show that the projection $\mathcal{Q}$ is non-expansive, which implies its image has a covering number no larger than the original space.

**Theorem C.7.** *The space $\Pi_{eq}$ has a covering number less than or equal to that of $\Pi$. Let $\mathcal{N}_{\infty,1}(\mathcal{F}, r)$ be the covering number of a function space $\mathcal{F}$ under the $l_{\infty,1}$-distance. Then, $\mathcal{N}_{\infty,1}(\Pi_{eq}, r) \le \mathcal{N}_{\infty,1}(\Pi, r)$.*

*Proof.* We show that $\mathcal{Q}$ is non-expansive. The $l_{\infty,1}$-distance between two policies $\pi_\phi$ and $\pi_\theta$ is

$$
d(\pi_\phi, \pi_\theta) = \sup_s \|\pi_\phi(s) - \pi_\theta(s)\|_1.
$$

The distance between their projections is:

$$d(\mathcal{Q}(\pi_\phi), \mathcal{Q}(\pi_\theta)) = \sup_s \left\| \tfrac{1}{2}\big(\pi_\phi(s) + K_g(\pi_\phi(L_g(s)))\big) - \tfrac{1}{2}\big(\pi_\theta(s) + K_g(\pi_\theta(L_g(s)))\big) \right\|_1$$

$$= \tfrac{1}{2} \sup_s \left\| (\pi_\phi(s) - \pi_\theta(s)) + K_g(\pi_\phi(L_g(s)) - \pi_\theta(L_g(s))) \right\|_1.$$

$$\leq \tfrac{1}{2} \sup_s \left( \|\pi_\phi(s) - \pi_\theta(s)\|_1 + \|K_g(\pi_\phi(L_g(s)) - \pi_\theta(L_g(s)))\|_1 \right).$$

$$\leq \tfrac{1}{2} \left( \sup_s \|\pi_\phi(s) - \pi_\theta(s)\|_1 + \sup_s \|\pi_\phi(L_g(s)) - \pi_\theta(L_g(s))\|_1 \right).$$

$$= \tfrac{1}{2} \big( d(\pi_\phi, \pi_\theta) + d(\pi_\phi, \pi_\theta) \big) = d(\pi_\phi, \pi_\theta),$$

where we use the triangle inequality, the fact that $K_g$ is a norm-preserving isometry, $\|K_g(a)\|_1 = \|a\|_1$, and that $L_g$ is a bijection, which implies that the supremum over $s$ equals the supremum over $L_g(s)$. Hence $\mathcal{Q}$ is non-expansive, and a non-expansive surjective map cannot increase the covering number. Following Lemma C.6, $\mathcal{N}(\Pi_{\mathrm{eq}}, r) \leq \mathcal{N}(\Pi, r)$. $\qquad\square$

The following lemma links coverings of the policy class (with metric $d$) to coverings of the induced return class (supremum over trajectories). This is the deterministic Lipschitz step that makes the entropy of returns comparable to the entropy of the policy class.

**Lemma C.8.** *For any policy set $\mathcal{P} \subseteq \Pi$ and any $\varepsilon > 0$,*

$$\mathcal{N}_{\infty,1}\big(\{\tau \mapsto R(\pi; \tau) : \pi \in \mathcal{P}\}, \varepsilon\big) \leq \mathcal{N}_{\infty,1}\big(\mathcal{P}, \varepsilon/L_R\big),$$

*where the left covering number is with respect to the sup-norm over trajectories and the right is with respect to $d(\cdot, \cdot)$.*

*Proof.* Let $\{\pi_1, \ldots, \pi_M\}$ be an $\varepsilon/L_R$-cover of $\mathcal{P}$ under $d(\cdot, \cdot)$. For any $\pi \in \mathcal{P}$ choose $j$ with $d(\pi, \pi_j) \leq \varepsilon/L_R$. Then for every trajectory $\tau$,

$$|R(\pi; \tau) - R(\pi_j; \tau)| \leq L_R d(\pi, \pi_j) \leq \varepsilon,$$

so the set $\{\tau \mapsto R(\pi_j; \tau)\}_{j=1}^M$ is an $\varepsilon$-cover of the return-class. Thus, the covering inequality holds. $\qquad\square$

## C.5 GENERALISATION OF REFLECTION-EQUIVARIANT SUBSPACE

We now prove a high-probability uniform bound over the equivariant class.

**Theorem C.9.** *With $\mathcal{R}_{\Pi_{\mathrm{eq}}} = \{\tau \mapsto R(\pi; \tau) : \pi \in \Pi_{\mathrm{eq}}\}$, fix any accuracy parameter $r \in (0, B)$ and confidence $\delta \in (0, 1)$. Then with probability at least $1 - \delta$,*

$$\sup_{\pi \in \Pi_{\mathrm{eq}}} |J(\pi) - \hat{J}_N(\pi)| \leq C \left( \int_r^B \sqrt{\frac{\log \mathcal{N}_{\infty,1}(\mathcal{R}_{\Pi_{\mathrm{eq}}}, \varepsilon)}{N}} d\varepsilon \right) + \frac{8r}{\sqrt{N}} + B\sqrt{\frac{\log(2/\delta)}{2N}},$$

*where $C$ is an absolute numeric constant, $J(\pi)$ is the population expected return and $\hat{J}_N(\pi) = \frac{1}{N} \sum_{i=1}^N R(\pi; \tau_i)$ is the empirical return on $N$ i.i.d. episodes $\tau_1, \ldots, \tau_N$.*

*Proof.* Let $\mathcal{F} = \mathcal{R}_{\Pi_{\mathrm{eq}}}$. Following Corollary 5.2, we have:

$$\mathbb{E}\Big[ \sup_{f \in \mathcal{R}_{\Pi_{\mathrm{eq}}}} \Big| \frac{1}{N} \sum_{i=1}^N (f(\tau_i) - \mathbb{E}[f]) \Big| \Big] \leq 2\mathbb{E}\big[\mathfrak{R}_N(\mathcal{R}_{\Pi_{\mathrm{eq}}})\big].$$

Applying Lemma C.1, for any $r > 0$:

$$\mathbb{E}\Big[ \sup_{f \in \mathcal{R}_{\Pi_{\mathrm{eq}}}} \Big| \frac{1}{N} \sum_{i=1}^N (f(\tau_i) - \mathbb{E}[f]) \Big| \Big] \leq C \left( \int_r^B \sqrt{\frac{\log \mathcal{N}_{\infty,1}(\mathcal{R}_{\Pi_{\mathrm{eq}}}, \varepsilon)}{N}} d\varepsilon \right) + \frac{8r}{\sqrt{N}}. \qquad (7)$$

Now apply Lemma C.2 to convert the expectation bound into a high-probability statement, with probability at least $1 - \delta$:

$$\sup_{f \in \mathcal{R}_{\Pi_{eq}}} \left| \frac{1}{N} \sum_{i=1}^{N} (f(\tau_i) - \mathbb{E}[f]) \right| \leq \mathbb{E} \left[ \sup_{f \in \mathcal{R}_{\Pi_{eq}}} \left| \frac{1}{N} \sum_{i=1}^{N} (f(\tau_i) - \mathbb{E}[f]) \right| \right] + B \sqrt{\frac{\log(2/\delta)}{2N}}. \quad (8)$$

Combining Equations 7 and 8 yields the claimed inequality. $\qquad\square$

### C.6 GENERALISATISABILITY OF PRISM

**Lemma C.10.** *If a policy $\pi$ satisfies $\mathcal{L}_{eq} \leq \varepsilon_{eq}$, then*

$$\sup_{s} \|\Delta_\pi(s)\|_1 \leq \sqrt{\frac{\varepsilon_{eq}}{p_{\min}}}.$$

*Consequently, the sup–$\ell_1$ distance between $\pi$ and its orbit projection $Q(\pi)$ satisfies*

$$d(\pi, Q(\pi)) = \sup_{s} \|\pi(s) - Q(\pi)(s)\|_1 \leq \sqrt{\frac{\varepsilon_{eq}}{p_{\min}}}.$$

*Proof.* Assume the state space has density $\frac{d\mu}{ds}(s) \geq p_{\min}$ on the common support. Let $s^*$ be such that $\|\Delta_\pi(s^*)\|_1 = \sup_s \|\Delta_\pi(s)\|_1$. The expectation is:

$$\varepsilon_{eq} = \mathbb{E}_\mu \left[ \|\Delta_\pi(s)\|_1^2 \right] = \int \|\Delta_\pi(s)\|_1^2 d\mu(s).$$

For any neighbourhood $B_\delta(s^*)$ of $s^*$:

$$\varepsilon_{eq} \geq \int_{B_\delta(s^*)} \|\Delta_\pi(s)\|_1^2 d\mu(s).$$

By continuity of $\|\Delta_\pi(\cdot)\|_1$ and the density lower bound:

$$\int_{B_\delta(s^*)} \|\Delta_\pi(s)\|_1^2 d\mu(s) \geq (\|\Delta_\pi(s^*)\|_1 - \epsilon)^2 \int_{B_\delta(s^*)} d\mu(s) \geq (\|\Delta_\pi(s^*)\|_1 - \epsilon)^2 p_{\min} \cdot \text{vol}(B_\delta(s^*)),$$

for sufficiently small $\delta$ and any $\epsilon > 0$. Taking $\delta \to 0$ and $\epsilon \to 0$:

$$\varepsilon_{eq} \geq p_{\min} \left( \sup_s \|\Delta_\pi(s)\|_1 \right)^2.$$

Rearranging gives $\sup_s \|\Delta_\pi(s)\|_1 \leq \sqrt{\frac{\varepsilon_{eq}}{p_{\min}}}$. $\qquad\square$

We can now translate this approximation to a bound on returns and to a covering-number statement.

**Theorem C.11.** *Let $\xi := \frac{1}{2}\sqrt{\varepsilon_{eq}/p_{\min}}$. Then for every policy $\pi$,*

$$|J(\pi) - J(Q(\pi))| \leq L_R \cdot d(\pi, Q(\pi)) \leq L_R \xi.$$

*Define the approximately reflection-equivariant class $\Pi_{approx}(\varepsilon_{eq}) := \{\pi \in \Pi : L_{eq}(\pi) \leq \varepsilon_{eq}\}$. Then every $\pi \in \Pi_{approx}(\varepsilon_{eq})$ lies in the sup-ball of radius $\xi$ around $\Pi_{eq}$. Consequently, for any target covering radius $r > \xi$:*

$$N_{\infty,1}\big(\Pi_{approx}(\varepsilon_{eq}), r\big) \leq N_{\infty,1}\big(\Pi_{eq}, r - \xi\big).$$

*Proof.* The first claim is that $|J(\pi) - J(Q(\pi))| \leq L_R \cdot d(\pi, Q(\pi)) \leq L_R \xi$.

First, we establish the $L_R$-Lipschitz property of the expected return $J(\pi) = \mathbb{E}_\tau[R(\pi; \tau)]$. Using the property from that the return function $R$ is $L_R$-Lipschitz, we have:

$$\begin{aligned} |J(\pi) - J(Q(\pi))| &= |\mathbb{E}_\tau[R(\pi; \tau) - R(Q(\pi); \tau)]| \\ &\leq \mathbb{E}_\tau \big[ |R(\pi; \tau) - R(Q(\pi); \tau)| \big] \\ &\leq \mathbb{E}_\tau \big[ L_R \cdot d(\pi, Q(\pi)) \big] = L_R \cdot d(\pi, Q(\pi)). \end{aligned}$$

Next, we bound the distance $d(\pi, Q(\pi))$. Using the definition of the projection $Q(\pi)$, we find the distance from $\pi$ to its projection:

$$d(\pi, Q(\pi)) = \sup_s \|\pi(s) - Q(\pi)(s)\|_1$$

$$= \sup_s \left\|\pi(s) - \tfrac{1}{2}\left(\pi(s) + K_g(\pi(L_g(s)))\right)\right\|_1$$

$$= \tfrac{1}{2} \sup_s \|\pi(s) - K_g(\pi(L_g(s)))\|_1.$$

The term inside the norm is equal to the equivariance mismatch $\Delta_\pi(s') := \pi(L_g(s')) - K_g(\pi(s'))$ evaluated at $s' = L_g(s)$, since $L_g$ is an involution.

$$\Delta_\pi(L_g(s)) = \pi(L_g(L_g(s))) - K_g(\pi(L_g(s))) = \pi(s) - K_g(\pi(L_g(s))).$$

Since $L_g$ is a bijection, $\sup_s \|\Delta_\pi(L_g(s))\|_1 = \sup_{s'} \|\Delta_\pi(s')\|_1$. By Lemma C.10, this supremum is bounded by $\xi$. Therefore:

$$d(\pi, Q(\pi)) = \frac{1}{2} \sup_{s'} \|\Delta_\pi(s')\|_1 \le \xi.$$

The second claim is that for any radius $r > \xi$, we have $N_{\infty,1}\big(\Pi_{approx}(\varepsilon_{eq}), r\big) \le N_{\infty,1}\big(\Pi_{eq}, r-\xi\big)$. We know that for any $\pi \in \Pi_{approx}(\varepsilon_{eq})$, its projection $Q(\pi) \in \Pi_{eq}$ satisfies $d(\pi, Q(\pi)) \le \xi$. This implies that the set $\Pi_{approx}(\varepsilon_{eq})$ is contained in a $\xi$-neighbourhood of $\Pi_{eq}$. Let $\{\pi_j\}_{j=1}^M$ be a minimal $(r - \xi)$-cover for $\Pi_{eq}$, where $M = N_{\infty,1}(\Pi_{eq}, r - \xi)$. Now, consider any policy $\pi \in \Pi_{approx}(\varepsilon_{eq})$. There must exist a centre $\pi_j$ from our cover such that $d(Q(\pi), \pi_j) \le r - \xi$. By the triangle inequality, we can bound the distance from $\pi$ to this centre $\pi_j$:

$$d(\pi, \pi_j) \le d(\pi, Q(\pi)) + d(Q(\pi), \pi_j)$$

$$\le \xi + (r - \xi) = r.$$

This shows that the set $\{\pi_j\}_{j=1}^M$ is an $r$-cover for $\Pi_{approx}(\varepsilon_{eq})$. Since we have found a valid cover of size $M$, the size of the minimal cover must be no larger:

$$N_{\infty,1}\big(\Pi_{approx}(\varepsilon_{eq}), r\big) \le N_{\infty,1}\big(\Pi_{eq}, r - \xi\big).$$

$\square$

**Theorem C.12.** *With $\mathcal{R}_{\Pi_{eq}} = \{\tau \mapsto R(\pi; \tau) : \pi \in \Pi_{eq}\}$, fix any accuracy parameter $r \in (0, B)$ and confidence $\delta \in (0, 1)$. Then with probability at least $1 - \delta$,*

$$\sup_{\pi \in \Pi_{approx}(\varepsilon_{eq})} |J(\pi) - \hat{J}_N(\pi)| \le C\left(\int_r^B \sqrt{\frac{\log \mathcal{N}_{\infty,1}(\mathcal{R}_{\Pi_{eq}}, \varepsilon)}{N}} d\varepsilon\right) + \frac{8r}{\sqrt{N}} + B\sqrt{\frac{\log(2/\delta)}{2N}} + 2L_R\xi.$$

*Proof.* For any policy $\pi \in \Pi_{approx}(\varepsilon_{eq})$, we can decompose the generalisation error using the triangle inequality by introducing its exact-equivariant projection $Q(\pi) \in \Pi_{eq}$:

$$|J(\pi) - \hat{J}_N(\pi)| \le |J(\pi) - J(Q(\pi))| + |J(Q(\pi)) - \hat{J}_N(Q(\pi))| + |\hat{J}_N(Q(\pi)) - \hat{J}_N(\pi)|.$$

We bound each of the three terms on the right-hand side.

From Theorem C.11, we have:

$$|J(\pi) - J(Q(\pi))| \le L_R \cdot d(\pi, Q(\pi)) \le L_R\xi.$$

Since the return function $R(\cdot; \tau)$ is $L_R$-Lipschitz:

$$|\hat{J}_N(Q(\pi)) - \hat{J}_N(\pi)| = \left|\frac{1}{N} \sum_{i=1}^N (R(Q(\pi); \tau_i) - R(\pi; \tau_i))\right|$$

$$\le \frac{1}{N} \sum_{i=1}^N |R(Q(\pi); \tau_i) - R(\pi; \tau_i)|$$

$$\le \frac{1}{N} \sum_{i=1}^N L_R \cdot d(\pi, Q(\pi)) \le L_R\xi.$$

The middle term, $|J(Q(\pi)) - \hat{J}_N(Q(\pi))|$, is the generalisation error for an exactly equivariant policy. Combining the bounds, we get:

$$\sup_{\pi \in \Pi_{approx}(\varepsilon_{eq})} |J(\pi) - \hat{J}_N(\pi)| \leq \sup_{\pi' \in \Pi_{eq}} |J(\pi') - \hat{J}_N(\pi')| + L_R \gamma.$$

Applying the high-probability bound from Theorem C.9 to the supremum over $\Pi_{eq}$ yields the final result. $\square$

## D  ADDITIONAL DETAILS OF ENVIRONMENTS

This appendix presents the tables on the environments and how the state space is divided into a symmetric and an asymmetric part. First Table 3 highlights the differences between environments in dimension sizes. Tables 4, 5, 6, and 7 show the division for mo-hopper-v5, mo-walker2d-v5, mo-halfcheetah-v5, and mo-swimmer-v5, respectively. The action space is always divided into an empty set for the asymmetric part, and the complete set for the symmetric part.

Table 3: Considered MuJoCo environments.

|  | State Space | Action Space | Reward Space |
|---|---|---|---|
| Mo-hopper-v5 | $\mathcal{S} \in \mathbb{R}^{11}$ | $\mathcal{A} \in \mathbb{R}^3$ | $\mathcal{R} \in \mathbb{R}^3$ |
| Mo-walker2d-v5 | $\mathcal{S} \in \mathbb{R}^{17}$ | $\mathcal{A} \in \mathbb{R}^6$ | $\mathcal{R} \in \mathbb{R}^2$ |
| Mo-halfcheetah-v5 | $\mathcal{S} \in \mathbb{R}^{17}$ | $\mathcal{A} \in \mathbb{R}^6$ | $\mathcal{R} \in \mathbb{R}^2$ |
| Mo-swimmer-v5 | $\mathcal{S} \in \mathbb{R}^8$ | $\mathcal{A} \in \mathbb{R}^2$ | $\mathcal{R} \in \mathbb{R}^2$ |

Table 4: Reflectional symmetry partition for mo-hopper-v5 observation space.

| Index | Observation Component | Type | Symmetry |
|---|---|---|---|
| 0 | z-coordinate of the torso | position | Asymmetric |
| 1 | angle of the torso | angle | Asymmetric |
| 2 | angle of the thigh joint | angle | Symmetric |
| 3 | angle of the leg joint | angle | Symmetric |
| 4 | angle of the foot joint | angle | Symmetric |
| 5 | velocity of the x-coordinate of the torso | velocity | Asymmetric |
| 6 | velocity of the z-coordinate of the torso | velocity | Asymmetric |
| 7 | angular velocity of the angle of the torso | angular velocity | Asymmetric |
| 8 | angular velocity of the thigh hinge | angular velocity | Symmetric |
| 9 | angular velocity of the leg hinge | angular velocity | Symmetric |
| 10 | angular velocity of the foot hinge | angular velocity | Symmetric |

Table 5: Reflectional symmetry partition for mo-walker2d-v5 observation space.

| Index | Observation Component | Type | Symmetry |
|---|---|---|---|
| 0 | z-coordinate of the torso | position | Asymmetric |
| 1 | angle of the torso | angle | Asymmetric |
| 2 | angle of the thigh joint | angle | Symmetric |
| 3 | angle of the leg joint | angle | Symmetric |
| 4 | angle of the foot joint | angle | Symmetric |
| 5 | angle of the left thigh joint | angle | Symmetric |
| 6 | angle of the left leg joint | angle | Symmetric |
| 7 | angle of the left foot joint | angle | Symmetric |
| 8 | velocity of the x-coordinate of the torso | velocity | Asymmetric |
| 9 | velocity of the z-coordinate of the torso | velocity | Asymmetric |
| 10 | angular velocity of the angle of the torso | angular velocity | Asymmetric |
| 11 | angular velocity of the thigh hinge | angular velocity | Symmetric |
| 12 | angular velocity of the leg hinge | angular velocity | Symmetric |
| 13 | angular velocity of the foot hinge | angular velocity | Symmetric |
| 14 | angular velocity of the left thigh hinge | angular velocity | Symmetric |
| 15 | angular velocity of the left leg hinge | angular velocity | Symmetric |
| 16 | angular velocity of the left foot hinge | angular velocity | Symmetric |

Table 6: Reflectional symmetry partition for mo-halfcheetah-v5 observation space.

| Index | Observation Component | Type | Symmetry |
|---|---|---|---|
| 0 | z-coordinate of the front tip | position | Asymmetric |
| 1 | angle of the front tip | angle | Asymmetric |
| 2 | angle of the back thigh | angle | Symmetric |
| 3 | angle of the back shin | angle | Symmetric |
| 4 | angle of the back foot | angle | Symmetric |
| 5 | angle of the front thigh | angle | Symmetric |
| 6 | angle of the front shin | angle | Symmetric |
| 7 | angle of the front foot | angle | Symmetric |
| 8 | velocity of the x-coordinate of front tip | velocity | Asymmetric |
| 9 | velocity of the z-coordinate of front tip | velocity | Asymmetric |
| 10 | angular velocity of the front tip | angular velocity | Asymmetric |
| 11 | angular velocity of the back thigh | angular velocity | Symmetric |
| 12 | angular velocity of the back shin | angular velocity | Symmetric |
| 13 | angular velocity of the back foot | angular velocity | Symmetric |
| 14 | angular velocity of the front thigh | angular velocity | Symmetric |
| 15 | angular velocity of the front shin | angular velocity | Symmetric |
| 16 | angular velocity of the front foot | angular velocity | Symmetric |

Table 7: Reflectional symmetry partition for mo-swimmer-v5 observation space.

| Index | Observation Component | Type | Symmetry |
|---|---|---|---|
| 0 | angle of the front tip | angle | Asymmetric |
| 1 | angle of the first rotor | angle | Symmetric |
| 2 | angle of the second rotor | angle | Symmetric |
| 3 | velocity of the tip along the x-axis | velocity | Asymmetric |
| 4 | velocity of the tip along the y-axis | velocity | Symmetric |
| 5 | angular velocity of the front tip | angular velocity | Asymmetric |
| 6 | angular velocity of first rotor | angular velocity | Symmetric |
| 7 | angular velocity of second rotor | angular velocity | Symmetric |

# E    ADDITIONAL DETAILS OF EXPERIMENTAL SETTINGS

**Evaluation Measures.** For the approximated Pareto front, we consider three well-known metrics that investigate the extent of the approximated front.

First, we consider hypervolume (HV) (Fonseca et al., 2006), which measures the volume of the objective space dominated by the approximated Pareto front relative to a reference point. A downside of many evaluation measures is that they require domain knowledge about the true underlying Pareto front, whereas HV only considers a reference point without any a priori knowledge, making it ideal to assess the volume of the front. The reference point is typically set to the nadir point or slightly worse, and following Felten et al. (2023), we set it to $-100$ for all objectives and environments. The HV is defined as follows:

$$HV(CS, \boldsymbol{r}) = \lambda \left( \bigcup_{\boldsymbol{cs} \in CS} \boldsymbol{x} \in \mathbb{R}^L : \boldsymbol{cs} \preceq \boldsymbol{x} \preceq \boldsymbol{r} \right),$$

where $CS = \boldsymbol{cs}_1, \boldsymbol{cs}_2, \ldots, \boldsymbol{cs}_n$ is the coverage set, or the Pareto front approximation, $\boldsymbol{r} \in \mathbb{R}^L$ is the reference point, $\boldsymbol{cs} \preceq \boldsymbol{x}$ means $cs_i \leq x_i$ for all objectives $i = 1, \ldots, L$, and $\lambda(\cdot)$ denotes the Lebesgue measure. Yet, hypervolume values are difficult to interpret, as they do not have a direct link to any notion of value or utility (Hayes et al., 2022).

As such, we also consider the Expected Utility Metric (EUM) (Zintgraf et al., 2015), which computes the expected maximum utility across different preference weight vectors, and is defined as follows:

$$EUM(CS, \mathcal{W}) = \frac{1}{|\mathcal{W}|} \sum_{\boldsymbol{\omega} \in \mathcal{W}} \max_{\boldsymbol{cs} \in CS} U(\boldsymbol{\omega}, \boldsymbol{cs}),$$

where $\mathcal{W} = \{\boldsymbol{\omega}_1, \boldsymbol{\omega}_2, \ldots, \boldsymbol{\omega}_k\}$ is a set of weight vectors, $|\mathcal{W}|$ is the cardinality of the weight set, $U(\boldsymbol{\omega}, \boldsymbol{cs})$ is the utility function, which is set to $U(\boldsymbol{\omega}, \boldsymbol{s}) = \boldsymbol{\omega} \cdot \boldsymbol{cs} = \sum_{i=1}^L \omega_i \cdot cs_i$.

To specifically assess performance with respect to distributional preferences, we also consider one metric designed to evaluate the optimality of the entire return distribution associated with the learned policies (Cai et al., 2023).

To be precise, we consider the Variance Objective (VO), which evaluates how well the policy set can balance the trade-off between maximising expected returns and minimising their variance. A set of $M$ random preference vectors is generated, where each vector specifies a different weighting between the expected return and its standard deviation for each objective. The satisfaction score $u(p_i, \pi_j)$ for a policy $\pi_j$ under preference $p_i$ is a weighted sum of the expected return $\mathbb{E}[Z(\pi_j)]$ and the negative standard deviation $-\sqrt{\mathrm{Var}[Z(\pi_j)]}$. The final metric is the mean score over these preferences, rewarding policies that achieve high expected returns with low variance:

$$\mathrm{VO}(\Pi, \{p_i\}_{i=1}^M) = \frac{1}{M} \sum_{i=1}^M \max_{\pi_j \in \Pi} u(p_i, \pi_j).$$

**Hyperparameters.** Due to time, computational limitations, and the excessive number of hyperparameters, we do not perform an extensive hyperparameter tuning process. Below are the used hyperparameters. All hyperparameters that are not mentioned below are set to their default value.

The probability of releasing sparse rewards $p_{\mathrm{rel}}$ is always set to a one-hot vector, where sparsity is imposed on the reward dimension related to moving forward. Since the main goal is to move forward, imposing sparsity on this channel should make it a more difficult task for the reward shaping model. Furthermore, we deal with extreme heterogeneous sparsity, where most channels exhibit regular rewards, but one channel only releases a reward at the end of an episode, making it more difficult for the model to link certain states and actions to the observed cumulative reward.

The hyperparameters in Table 8 for ReSymNet are identical for each environment. The advantage of using the same hyperparameters for each environment is that if one configuration performs well everywhere, it could indicate that the proposed method is inherently stable, especially given the noted diversity between the considered environments. However, this does come at a cost of potentially suboptimal performance per environment.

Table 8: Hyperparameters for ReSymNet.

|  | PRISM |
|---|---|
| Initial collection $N$ | 1000 |
| Expert collection $E$ | 1000 |
| Number of refinements $IR$ | 2 |
| Timesteps per cycle $M$ | 100,000 |
| Epochs | 1000 |
| Learning rate | 0.005 |
| Learning rate scheduler | Exponential |
| Learning rate decay | 0.99 |
| Ensemble size $|\mathcal{E}|$ | 3 |
| Hidden dimension | 256 |
| Dropout | 0.3 |
| Initialisation | Kaiman (He et al., 2015) |
| Validation split | 0.2 |
| Patience | 20 |
| Batch size | 32 |

The hyperparameter controlling the symmetry loss differs per environment, since some environments require strict equivariance, whereas others require a more flexible approach. Table 9 shows the used values.

Table 9: SymReg hyperparameter.

|  | Mo-hopper-v5 | Mo-walker2d-v5 | Mo-halfcheetah-v5 | Mo-swimmer-v5 |
|---|---|---|---|---|
| $\lambda$ | 0.01 | 1 | 0.01 | 0.005 |

## F  PARETO FRONTS

Figure 6 shows the approximated Pareto fronts. The results demonstrate that shaped rewards yield superior performance, covering a wider and more optimal range of the objective space compared to dense and sparse rewards.

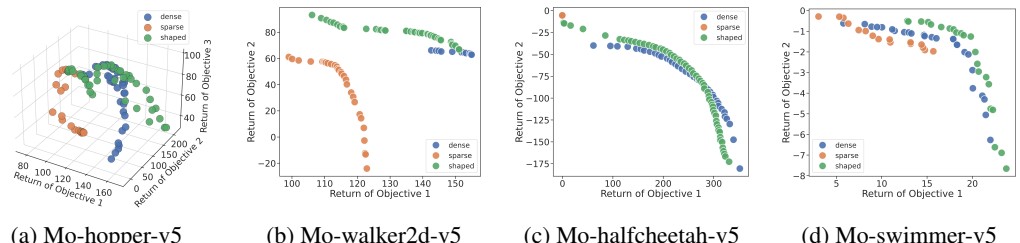

(a) Mo-hopper-v5  (b) Mo-walker2d-v5  (c) Mo-halfcheetah-v5  (d) Mo-swimmer-v5

Figure 6: The approximated Pareto front for dense rewards (blue dots), sparse rewards (orange dots), and shaped rewards (green dots). Sparsity is imposed on the first reward objective.

## G  ABLATION STUDY

Tables 10 and 11 report the obtained values for the ablation study. Results are again averaged over ten trials, similar to the main experiments.

Table 10: PRISM ablation study results. We report the average hypervolume (HV), Expected Utility Metric (EUM), and Variance Objective (VO) over 10 trials, with the standard error shown in grey. w/o is the abbreviation of without. The largest values are in bold font.

| Environment | Metric | PRISM | w/o residual | w/o dense rewards | w/o ensemble | w/o refinement | w/o loss |
|---|---|---|---|---|---|---|---|
| Mo-hopper-v5 | HV ($\times 10^7$) | **1.58** $\pm$ 0.05 | 1.29 $\pm$ 0.09 | 1.38 $\pm$ 0.11 | 1.38 $\pm$ 0.08 | 1.55 $\pm$ 0.04 | 1.42 $\pm$ 0.07 |
| | EUM | **147.43** $\pm$ 2.61 | 128.40 $\pm$ 6.06 | 134.67 $\pm$ 6.89 | 135.28 $\pm$ 4.91 | 145.89 $\pm$ 2.73 | 137.85 $\pm$ 4.22 |
| | VO | **66.66** $\pm$ 1.40 | 58.61 $\pm$ 2.71 | 61.21 $\pm$ 3.03 | 61.51 $\pm$ 2.19 | 66.54 $\pm$ 1.34 | 62.71 $\pm$ 1.83 |
| Mo-walker2d-v5 | HV ($\times 10^4$) | **4.77** $\pm$ 0.07 | 4.65 $\pm$ 0.11 | 4.66 $\pm$ 0.06 | 4.60 $\pm$ 0.08 | 4.60 $\pm$ 0.09 | 4.58 $\pm$ 0.13 |
| | EUM | **120.43** $\pm$ 1.64 | 114.33 $\pm$ 2.48 | 116.83 $\pm$ 1.65 | 113.79 $\pm$ 2.02 | 114.98 $\pm$ 2.84 | 112.77 $\pm$ 3.01 |
| | VO | **59.35** $\pm$ 0.80 | 56.46 $\pm$ 1.21 | 57.67 $\pm$ 0.73 | 56.19 $\pm$ 0.97 | 57.03 $\pm$ 1.42 | 55.59 $\pm$ 1.44 |
| Mo-halfcheetah-v5 | HV ($\times 10^4$) | **2.25** $\pm$ 0.18 | 1.95 $\pm$ 0.20 | 2.08 $\pm$ 0.21 | 1.91 $\pm$ 0.19 | 2.23 $\pm$ 0.18 | 1.90 $\pm$ 0.19 |
| | EUM | 89.94 $\pm$ 15.33 | 73.06 $\pm$ 16.57 | 82.24 $\pm$ 16.97 | 81.60 $\pm$ 17.65 | **92.68** $\pm$ 14.79 | 71.12 $\pm$ 16.91 |
| | VO | 40.72 $\pm$ 7.02 | 32.99 $\pm$ 7.65 | 37.31 $\pm$ 7.99 | 36.76 $\pm$ 8.06 | **42.28** $\pm$ 6.85 | 32.12 $\pm$ 7.75 |
| Mo-swimmer-v5 | HV ($\times 10^4$) | **1.21** $\pm$ 0.00 | **1.21** $\pm$ 0.00 | 1.20 $\pm$ 0.00 | 1.20 $\pm$ 0.00 | **1.21** $\pm$ 0.00 | 1.20 $\pm$ 0.00 |
| | EUM | 9.44 $\pm$ 0.14 | 9.39 $\pm$ 0.15 | 9.07 $\pm$ 0.11 | 9.25 $\pm$ 0.13 | **9.46** $\pm$ 0.13 | 9.35 $\pm$ 0.14 |
| | VO | **4.24** $\pm$ 0.07 | 4.20 $\pm$ 0.08 | 4.09 $\pm$ 0.05 | 4.15 $\pm$ 0.08 | **4.24** $\pm$ 0.07 | **4.24** $\pm$ 0.07 |

Table 11: ReSymNet ablation study results. We report the average hypervolume (HV), Expected Utility Metric (EUM), and Variance Objective (VO) over 10 trials, with the standard error shown in grey. w/o is the abbreviation of without.

| Environment | Metric | uniform | random |
|---|---|---|---|
| Mo-hopper-v5 | HV ($\times 10^7$) | 1.38 $\pm$ 0.08 | 0.49 $\pm$ 0.06 |
| | EUM | 135.19 $\pm$ 5.30 | 65.22 $\pm$ 6.63 |
| | VO | 63.90 $\pm$ 2.34 | 29.62 $\pm$ 3.68 |
| Mo-walker2d-v5 | HV ($\times 10^4$) | 4.67 $\pm$ 0.07 | 1.18 $\pm$ 0.10 |
| | EUM | 116.72 $\pm$ 2.11 | 16.52 $\pm$ 4.98 |
| | VO | 56.22 $\pm$ 1.01 | 3.77 $\pm$ 2.46 |
| Mo-halfcheetah-v5 | HV ($\times 10^4$) | 0.98 $\pm$ 0.00 | 0.78 $\pm$ 0.05 |
| | EUM | -1.34 $\pm$ 0.39 | -10.52 $\pm$ 2.67 |
| | VO | -0.85 $\pm$ 0.20 | -6.51 $\pm$ 1.48 |
| Mo-swimmer-v5 | HV ($\times 10^4$) | 1.09 $\pm$ 0.01 | 1.10 $\pm$ 0.02 |
| | EUM | 4.37 $\pm$ 0.69 | 3.75 $\pm$ 0.87 |
| | VO | 1.56 $\pm$ 0.33 | 1.06 $\pm$ 0.40 |

# H  GENERALISABILITY

## H.1  SPARSITY ON OTHER OBJECTIVES

We further investigate the robustness of PRISM by inverting the sparsity setting: we maintain the forward velocity reward as dense but make the control cost objective sparse. Table 12 shows that, without hyperparameter tuning, PRISM handles this problem much better than the baselines.

For mo-hopper-v5, PRISM improves HV by 16% over the oracle ($1.51 \times 10^7$ compared to $1.30 \times 10^7$) and 27% over the baseline. Similar gains are observed for mo-walker2d-v5, where PRISM achieves a 9% HV improvement over the oracle and 45% over the baseline. Notably, in mo-halfcheetah-v5, the baseline suffers a collapse (HV of 0.00), whereas PRISM recovers the performance to exceed the oracle ($1.72 \times 10^4$ against $1.70 \times 10^4$). These improvements imply that PRISM effectively reconstructs the dense penalty signal, preventing the agent from exploiting the delay to maximise velocity at the cost of extreme energy inefficiency.

Improvements in EUM follow the same trend, with mo-walker2d-v5 showing an increase of roughly 33% compared to the baseline (114.62 vs 85.95). On distributional metrics, PRISM delivers more consistent performance than the baseline. In mo-swimmer-v5, the baseline's VO drops to $-0.61$, indicating high instability, whereas PRISM achieves 3.95, comparable to the oracle (4.22). These gains are crucial because they indicate that PRISM produces Pareto fronts that are not only high-performing but also balanced and robust, effectively mitigating the high-variance behaviour from the baseline.

Table 12: Experimental results on the control cost objective. We report the average hypervolume (HV), Expected Utility Metric (EUM), and Variance Objective (VO) over 10 trials, with the standard error shown in grey. The largest (best) values are in bold font.

| Environment | Metric | Oracle | Baseline | PRISM |
|---|---|---|---|---|
| Mo-hopper-v5 | HV ($\times 10^7$) | $1.30 \pm 0.13$ | $1.19 \pm 0.10$ | $\mathbf{1.51} \pm 0.11$ |
| | EUM | $129.04 \pm 7.96$ | $124.82 \pm 7.21$ | $\mathbf{142.89} \pm 7.38$ |
| | VO | $59.07 \pm 3.45$ | $56.21 \pm 3.20$ | $\mathbf{67.58} \pm 3.31$ |
| Mo-walker2d-v5 | HV ($\times 10^4$) | $4.21 \pm 0.11$ | $3.16 \pm 0.13$ | $\mathbf{4.59} \pm 0.14$ |
| | EUM | $107.58 \pm 2.86$ | $85.95 \pm 3.27$ | $\mathbf{114.62} \pm 2.80$ |
| | VO | $53.22 \pm 1.39$ | $41.29 \pm 1.49$ | $\mathbf{54.84} \pm 1.25$ |
| Mo-halfcheetah-v5 | HV ($\times 10^4$) | $1.70 \pm 0.20$ | $0.00 \pm 0.00$ | $\mathbf{1.72} \pm 0.19$ |
| | EUM | $\mathbf{81.29} \pm 21.85$ | $-101.49 \pm 3.23$ | $76.50 \pm 20.85$ |
| | VO | $\mathbf{36.84} \pm 10.06$ | $-56.26 \pm 1.63$ | $31.27 \pm 8.68$ |
| Mo-swimmer-v5 | HV ($\times 10^4$) | $\mathbf{1.21} \pm 0.00$ | $1.05 \pm 0.02$ | $\mathbf{1.21} \pm 0.01$ |
| | EUM | $\mathbf{9.41} \pm 0.12$ | $1.50 \pm 1.00$ | $9.32 \pm 0.19$ |
| | VO | $\mathbf{4.22} \pm 0.08$ | $-0.61 \pm 0.68$ | $3.95 \pm 0.08$ |

## H.2 SENSITIVITY TO SPARSITY

Figure 7 demonstrates that PRISM maintains robust performance across varying levels of reward sparsity. While performance is generally consistent, we observe minor fluctuations at intermediate values (e.g., $p_{rel} = 0.2$ in mo-hopper-v5 and mo-walker2d-v5). Two key factors explain this behaviour: (1) PRISM was hyperparameter-tuned specifically for the extreme sparsity setting ($p_{rel} = 0$), which is the most challenging MORL scenario. We utilised a fixed set of hyperparameters across all experiments to demonstrate method stability rather than optimising for each sparsity level, and (2) increasing $p_{rel}$ increases the number of available reward signals (data points) per episode. Since ReSymNet was calibrated for the data-scarce sparse setting, the increase of supervision targets at higher $p_{rel}$ levels changes optimisation dynamics, leading to temporary instability. Despite these factors, PRISM consistently recovers high performance, proving its capability to handle heterogeneous reward structures without requiring specific tuning for denser environments.

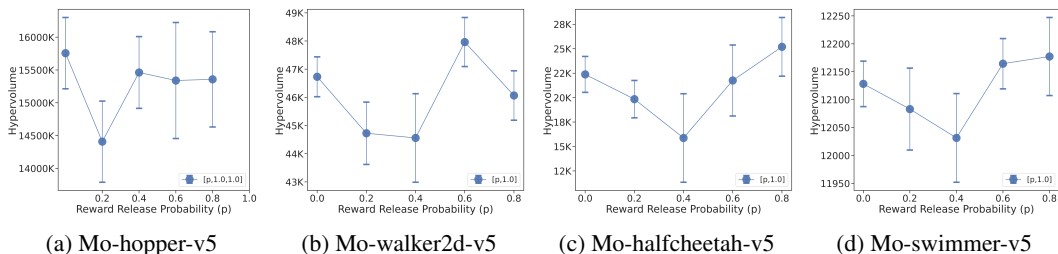

| (a) Mo-hopper-v5 | (b) Mo-walker2d-v5 | (c) Mo-halfcheetah-v5 | (d) Mo-swimmer-v5 |
|---|---|---|---|

Figure 7: The obtained hypervolume for various levels of sparsity for PRISM.

## H.3 SENSITIVITY TO MORL ALGORITHMS

To demonstrate that PRISM is a model-agnostic framework not limited to specific architectures, we evaluated its performance using GPI-PD (Generalised Policy Improvement with Linear Dynamics) (Alegre et al., 2023) as an alternative backbone to CAPQL. Table 13 confirms that PRISM remains highly effective, consistently outperforming the sparse baseline and obtaining near-oracle performance.

In mo-hopper-v5, PRISM achieves an HV of $1.65 \times 10^7$, matching the oracle exactly and far exceeding the baseline ($0.67 \times 10^7$). This trend of near-perfect recovery is consistent across mo-walker2d-v5 and mo-swimmer-v5. This indicates that the shaped rewards generated by ReSymNet

Table 13: Experimental results of GPI-PD. We report the average hypervolume (HV), Expected Utility Metric (EUM), and Variance Objective (VO) over 10 trials, with the standard error shown in grey. The largest (best) values are in bold font.

| Environment | Metric | Oracle | Baseline | PRISM |
|---|---|---|---|---|
| Mo-hopper-v5 | HV ($\times 10^7$) | **1.65** $\pm$ 0.10 | 0.67 $\pm$ 0.04 | **1.65** $\pm$ 0.07 |
| | EUM | **151.45** $\pm$ 5.87 | 85.87 $\pm$ 3.17 | 148.19 $\pm$ 4.26 |
| | VO | **72.26** $\pm$ 2.90 | 41.21 $\pm$ 1.44 | 70.24 $\pm$ 2.51 |
| Mo-walker2d-v5 | HV ($\times 10^4$) | **5.93** $\pm$ 0.10 | 3.20 $\pm$ 0.23 | 5.61 $\pm$ 0.10 |
| | EUM | **141.88** $\pm$ 2.38 | 76.41 $\pm$ 6.47 | 132.67 $\pm$ 2.26 |
| | VO | **67.63** $\pm$ 1.17 | 35.64 $\pm$ 3.91 | 63.19 $\pm$ 1.75 |
| Mo-halfcheetah-v5 | HV ($\times 10^4$) | 1.80 $\pm$ 0.22 | 1.00 $\pm$ 0.02 | **2.24** $\pm$ 0.16 |
| | EUM | **164.75** $\pm$ 14.21 | -1.31 $\pm$ 0.54 | 99.89 $\pm$ 8.06 |
| | VO | **73.90** $\pm$ 7.05 | -1.14 $\pm$ 0.31 | 40.74 $\pm$ 5.17 |
| Mo-swimmer-v5 | HV ($\times 10^4$) | **1.23** $\pm$ 0.01 | 1.12 $\pm$ 0.01 | 1.22 $\pm$ 0.00 |
| | EUM | **9.68** $\pm$ 0.17 | 5.17 $\pm$ 0.58 | 9.56 $\pm$ 0.13 |
| | VO | 4.23 $\pm$ 0.14 | 2.18 $\pm$ 0.39 | **4.37** $\pm$ 0.18 |

are robust enough to guide different policy optimisation mechanisms effectively. In mo-halfcheetah-v5, PRISM achieves a significantly higher HV (2.24) compared to the oracle (1.80).

Notably, these results were obtained with minimal hyperparameter tuning due to computational constraints. While this lack of fine-tuning explains the slight gap in EUM/VO metrics for mo-halfcheetah-v5 compared to the oracle, the method's ability to achieve such strong results with a completely different backbone highlights PRISM's inherent stability and generalisability.

# I Declaration on Large Language Models

Large Language Models (LLMs) were used for (1) polishing the wording of the manuscript for clarity and readability, (2) brainstorming about algorithm names and their abbreviations, and (3) searching for algorithms for consideration in the preliminary stage.

