# OpenReview forum: "Exploiting Reflectional Symmetry in Heterogeneous MORL"
_ICLR.cc/2026/Conference — Submitted to ICLR 2026_

### Official Review · Reviewer_pWip · 2025-10-28

**Soundness:** 3
**Presentation:** 3
**Contribution:** 2
**Rating:** 4
**Confidence:** 2

**Summary:**

This paper addresses heterogeneous Multi-Objective Reinforcement Learning (MORL), where different reward channels vary significantly in sparsity and magnitude, leading dense objectives to overshadow sparse but crucial long-term ones. The authors propose PRISM, a novel framework that enhances sample efficiency and generalisation in such settings. Theoretical analysis shows that PRISM constrains the policy search to a lower-complexity, reflection-equivariant subspace, leading to tighter generalisation bounds.

**Strengths:**

1. The paper introduces a fresh perspective on heterogeneous MORL, an underexplored yet realistic challenge. The combination of reward alignment through residual networks and symmetry-based inductive bias is novel and creative. Using reflectional symmetry as a regulariser for MORL agents is particularly original, bridging theoretical symmetry principles with multi-objective RL.
2. The work is thorough, combining rigorous theoretical analysis (covering-number and Rademacher-based generalisation bounds) with extensive experiments and ablation studies.
3. The proposed approach addresses a practical and fundamental problem in MORL—reward heterogeneity—which often arises in robotics and control. The demonstrated ability to outperform even dense-reward oracles suggests real impact for efficient multi-objective learning in sparse or imbalanced settings.

**Weaknesses:**

1. The reflectional symmetry assumption may hold for certain robotic domains (e.g., locomotion) but is less applicable to asymmetric or high-dimensional tasks. The method’s generality could be questioned—evaluating on more asymmetric tasks would clarify its broader utility.
2. ReSymNet relies on at least one dense reward signal to align sparse ones, which limits applicability in fully sparse or partially observable scenarios. A discussion or ablation on settings without dense channels would strengthen the claim of general applicability.
3. The notation is somewhat inconsistent and insufficiently defined throughout the paper, such as $D$, $r_t^{sp}$, and parameters like $\psi$ used in Eq. (1) are either not introduced clearly.
4. The problem is formulated as a multi-objective RL task, yet much of the theoretical and analytical development is effectively single-objective. The generalisation analysis, reflectional-equivariance projection, and covering-number proofs are all framed in terms of a scalar return rather than vector-valued multi-objective returns.

**Questions:**

See the Weaknesses.

---

> ### Author Response · Authors · 2025-11-19
> **Author Response to Reviewer pWip**
>
> We thank the reviewer for their constructive feedback.  All your concerns have been duly addressed below. We have also updated the manuscript accordingly. We sincerely hope that the reviewer can take into account the response we have made and reevaluate the merit of this paper.
>
> **Q1:** _The reflectional symmetry assumption may hold for certain robotic domains (e.g., locomotion) but is less applicable to asymmetric or high-dimensional tasks. The method’s generality could be questioned—evaluating on more asymmetric tasks would clarify its broader utility._
>
> **A1:** We respectfully clarify as follows. In this work, the symmetric and asymmetric components of the state and action spaces are indeed defined manually. However, the central contribution of our method is not in how these components are defined, but in how they are leveraged to improve learning efficiency and policy performance. Existing research already delves into the problem of automatic detection of symmetries, for example, by observing that symmetric state-action pairs often lead to statistically similar features [1].
>
> **Q2:** _ReSymNet relies on at least one dense reward signal to align sparse ones, which limits applicability in fully sparse or partially observable scenarios. A discussion or ablation on settings without dense channels would strengthen the claim of general applicability._
>
> **A2:** We perform an ablation study with results in the appendix, which includes the setting w/o dense rewards. This approach removes the dense signals as input and only uses the state and action as input to ReSymNet. The results still outperform the oracle in 3 out of 4 environments, indicating the robustness of the framework even without dense rewards.
>
> **Q3:** _The notation is somewhat inconsistent and insufficiently defined throughout the paper, such as, parameters like used in Eq. (1) are either not introduced clearly._
>
> **A3:** Thanks and addressed.
>
> **Q4:** _The problem is formulated as a multi-objective RL task, yet much of the theoretical and analytical development is effectively single-objective. The generalisation analysis, reflectional-equivariance projection, and covering-number proofs are all framed in terms of a scalar return rather than vector-valued multi-objective returns._
>
> **A4:** Thanks and addressed. We have extended the theory to cover vector-valued multi-objective returns, as the following corollary.
>
> **Corollary C.3.** Let $\Pi$ be a policy class equipped with a metric $d(\cdot,\cdot)$, and let $\mathbf{R}(\pi;\tau)\in\mathbb{R}^L$ denote the vector-valued return of policy $\pi$ on trajectory $\tau$. Following Assumption 1:
>
> $\sup_{\tau}||\mathbf{R}(\pi;\tau)-\mathbf{R}(\tilde\pi;\tau)||\_\infty \le L_R d(\pi,\tilde\pi) \qquad\text{for all }\pi,\tilde\pi\in\Pi.$
>
> For any weight vector $\omega\in\mathbb{R}^L$ define the scalarised return $ R^\omega(\pi;\tau)=\omega^\top \mathbf{R}(\pi;\tau)$ and let $\mathcal R\_{\Pi}^{\omega}$ be the class of scalarised returns induced by $\Pi$. Then for any $\varepsilon>0$,
>
> $\mathcal{N}\_{\infty,1}(\mathcal{R}\_{\Pi}^{\omega}, \varepsilon) \le \mathcal{N}\_{\infty,1}\big(\Pi, \varepsilon / L\_R^{\omega}\big), \qquad \text{where } L\_R^{\omega} := ||\omega||\_1L\_R.$
>
> In particular, when $||\omega||\_1 = 1$ we have $L\_R^{\omega}=L\_R$, and the scalarised return class has covering numbers no larger than those of the policy class. Consequently, any complexity reduction obtained by projecting $\Pi$ to an equivariant subspace (e.g., $\Pi{eq}$) is inherited by the scalarised objective class $\mathcal R\_{\Pi}^{\omega}$.
>
> The theoretical analysis was originally framed in terms of scalar returns; we extended this to MORL via scalarisation. Since Pareto-optimal policies in convex MORL can be characterised as maximisers of scalarised returns (as used in our backbone CAPQL) [2-3], proving generalisation for an arbitrary scalar return implies guarantees for the scalarised components of the Pareto front.
>
> Moreover, our original theoretical result, the reduction of hypothesis complexity via reflectional symmetry, is a property of the policy space, independent of the output reward dimension. By constraining the policy search to the equivariant subspace, we reduce the complexity of the function class itself, thereby improving sample efficiency for all vector objectives that respect the agent's physical symmetry.
>
> **References:**
>
> [1] Mahajan, A. and Tulabandhula, T. Symmetry learning for function approximation in reinforcement learning. arXiv preprint arXiv:1706.02999, 2017.
>
> [2] Roijers, D.M., Whiteson, S., Oliehoek, F.A.. Computing Convex Coverage Sets for Multi-objective Coordination Graphs, ADT, 2013.
>
> [3] Lu, H., Herman D., and Yu, Y. Multi-objective reinforcement learning: Convexity, stationarity and Pareto optimality, ICLR, 2023.

---

> > ### Comment · Reviewer_pWip · 2025-11-25
> >
> > Thank you for your response. After reading the comments from the other reviewers, I have decided to keep my score unchanged for now.

---

### Official Review · Reviewer_itAC · 2025-10-31

**Soundness:** 2
**Presentation:** 3
**Contribution:** 2
**Rating:** 2
**Confidence:** 4

**Summary:**

This paper proposes an algorithm called PRISM to address the overshadowing problem in heterogeneous multi-objective reinforcement learning (MORL), where some objectives are dense while others are sparse. The proposed architecture, ReSymNet, performs reward decomposition for sparse objectives using residual blocks. By learning pseudo-dense signals for these sparse objectives, the method enables policy training within a standard MORL framework. To enhance policy generalization, an auxiliary loss is incorporated into the policy loss to enforce reflectional equivariance. Experimental results demonstrate that integrating the proposed method into a standard MORL algorithm (specifically, CAPQL) improves performance according to several widely used MORL metrics.

**Strengths:**

- Provides theoretical support.

- Explores an interesting direction in MORL.

- Modular design: can be equipped with any (multi-policy) MORL algorithm.

**Weaknesses:**

1. Lack of comparison with reward decomposition methods in delayed-reward RL

- To my surprise, the paper does not adequately compare the proposed approach with existing reward decomposition methods [A,B,C,D], both in the literature review and in experiments. A simple extension (e.g., adding the dense reward component as an additional input) in [A,B,C,D] could make such experimental comparisons feasible in MORL settings. Comparison might even be possible without modification, since “state-action features already contain substantial signal (Line 465)”. The authors should review and discuss the relevant works including [A,B,C,D], clarifying whether the only difference lies in architectural choices.

 -  A. https://arxiv.org/pdf/2410.20176
    B. https://arxiv.org/pdf/2402.03771
    C. https://arxiv.org/pdf/2111.13485
    D. https://arxiv.org/pdf/2010.12718

2. Unclear motivation for SymReg and its connection to ReSymNet

- 2-1. The motivation for incorporating SymReg alongside ReSymNet is unclear, as they appear to represent orthogonal approaches. An ablation study combining ReSymNet with other well-known policy feature learning or regularization methods would help clarify the contribution.

- 2-2. The authors should also discuss why SymReg synergizes with the ReSymNet architecture, whereas other representation methods may not.

3. Manual decomposition of $s_{asym}$ and $s_{sym}$
​
- 3-1. It seems that $s_{asym}$ and $s_{sym}$ are manually decomposed for each environment, which raises concerns about the generality of the proposed method.
- 3-2. Can we also consider more general augmentation other than mirroring in the PRISM framework?

4. Lack of key experiments for ablation study

- 4-1. For sanity check, the authors should include (i) a uniform reward distribution (i.e., 1/T * last-reward for each timestep), and (ii) a random reward generation.

- 4-2. The paper should visualize the final 2D/3D Pareto fronts of each algorithm in addition to the quantitative results in Table 1 (similar to Fig. 4). Plus, using these visualizations, please discuss why the proposed method outperforms the oracle.

- 4-3. It would be helpful to qualitatively analyze the behavior of the learned reward signals, similar to Fig. 5 in reference [A].

- 4-4. Since the proposed method is modular, applying it to other MORL algorithms beyond CAPQL could further demonstrate its generalizability.

- 4-5. Testing in environments with different $p_{\text{rel}}$ values would also strengthen claims of generalization.

- 4-6. The first objective in each environment corresponds to the velocity component. It would be meaningful to investigate whether similar improvements occur when delaying other quantities, such as energy (i.e., the second objective).

5. Need for truly sparse reward settings

- It is interesting that the “w/o refinement” variant performs moderately well. This may be because non-zero non-trivial rewards are still provided at the end of episodes in delayed reward settings. To better reflect the truly sparse nature of certain objectives, the authors could consider other sparse environments in nature. For example, MO-Lunar-Lander [E], where the first objective represents the success or failure of each trajectory, could serve as a suitable testbed.

- E. https://openreview.net/forum?id=jfwRLudQyj

6. Additional issues

- How to split D train and D val in the algorithm?
- If the authors used MO-Gymnasium, please cite reference [E] in addition to the original MuJoCo citation.
- In the related work section, papers categorized as “(iii) meta-policy” could also be considered “multi-policy” in many MORL studies. Note that in the MORL community, the term meta-policy is sometimes reserved for works that explicitly adopt meta-learning approaches, such as [F].
- Minor: A left parenthesis is missing in Line 722.
- F. https://arxiv.org/pdf/1811.03376

**Questions:**

Please see the above weaknesses.

---

> ### Author Response · Authors · 2025-11-19
> **Author Response to Reviewer itAC (1/3)**
>
> We thank the reviewer for their constructive feedback.  All your concerns have been duly addressed below. We have also updated the manuscript accordingly. We sincerely hope that the reviewer can take into account the response we have made and reevaluate the merit of this paper.
>
> **Q1:** _Lack of comparison with reward decomposition methods in delayed-reward RL_
>
> _To my surprise, the paper does not adequately compare the proposed approach with existing reward decomposition methods [A,B,C,D], both in the literature review and in experiments. A simple extension (e.g., adding the dense reward component as an additional input) in [A,B,C,D] could make such experimental comparisons feasible in MORL settings. Comparison might even be possible without modification, since “state-action features already contain substantial signal (Line 465)”. The authors should review and discuss the relevant works including [A,B,C,D], clarifying whether the only difference lies in architectural choices._
>
> **A1:** Thanks. We will carefully cite and discuss these papers. We agree that methods like RRD, IRCR, RBT, and CoDeTr [A-D] could be architecturally extended and combined with SymReg. However, the critical distinction lies in the theoretical mechanism of the reward shaping itself. While [A-D] rely on statistical estimation (e.g., Monte-Carlo in RRD, uniform smoothing in IRCR) or sequence modelling (Transformers in RBT/CoDeTr) to redistribute scalar returns, ReSymNet is explicitly grounded in the theory of Scaled Opportunity Value (Laud, 2004).
>
> Further, as explained in the reward shaping paragraph of Sec 2, these methods, as single-objective reward shaping techniques, do not apply to MORL; extending them to MORL is non-trivial. We quoted here
>
> "These approaches improve sample efficiency in single-objective RL, but do not extend naturally to MORL, where heterogeneous sparsity and scale can distort learning dynamics and Pareto-optimal trade-offs. "
>
> **Q2:** _The motivation for incorporating SymReg alongside ReSymNet is unclear, as they appear to represent orthogonal approaches. An ablation study combining ReSymNet with other well-known policy feature learning or regularization methods would help clarify the contribution._
>
> _The authors should also discuss why SymReg synergizes with the ReSymNet architecture, whereas other representation methods may not._
>
> **A2:** We respectfully clarify as follows. Heterogeneous reward structures cause asymmetric policy learning, violating the agent's physical symmetry requirements. When dense objectives provide immediate gradients at every timestep while sparse objectives only signal at the episode end, the policy overfits to dense objectives in the regions encountered early in training. Even though the environment has reflectional symmetry, the learned policy fails to respect this symmetry because the temporal misalignment of heterogeneous rewards drives asymmetric exploration and learning dynamics. E.g., left leg vs. right leg in locomotion should have reflectional symmetry, but if left leg is explored more because of the heterogeneity in rewards, the symmetry is violated.
>
> In light of this vision, we propose a unified PRISM approach with two components that are integrated together: (1) ReSymNet, an efficient backbone reward shaping model to align heterogeneous reward channels, based on strong theoretical motivations, and (2) SymReg, a novel reflectional equivariance regulariser, to enforce reflectional symmetry in the reward shaping process.
>
> Both components address the reward heterogeneity / asymmetry problem directly. ReSymNet eliminates temporal heterogeneity by aligning all objectives to the same, dense frequency, ensuring all objectives provide equally informative training signals, and thus preventing asymmetric learning. SymReg prevents the policy learning from exploiting asymmetric, shortcut solutions that favour dense objectives, by enforcing reflectional equivariance across all objectives.
>
> This motivation was stated in the Abstract and Introduction. Following your question, we have re-emphasised it in the Introduction as follows,
>
> "The complementary components of PRISM synergise as follows. Heterogeneous reward structures cause asymmetric policy learning that violates the agent's physical symmetry: when dense objectives provide immediate gradients while sparse objectives only signal at the end of an episode, the policy may overfit to the denser objectives in specific states, failing to respect reflectional symmetry. ReSymNet eliminates temporal heterogeneity by aligning objectives to the same frequency, whereas SymReg enforces reflectional symmetry by preventing asymmetric learning dynamics."
>
> Further, we would love to note - having more than one contribution in one paper is never a fault. There have been too many important papers in the history of machine learning that present two or more contributions in their papers; for example, see [1-6].

---

> ### Author Response · Authors · 2025-11-19
> **Author Response to Reviewer itAC (2/3)**
>
> **Q3.1:** _It seems that s_asym and s_sym are manually decomposed for each environment, which raises concerns about the generality of the proposed method._
>
> **A3.1:** In this work, the symmetric and asymmetric components of the state and action spaces are indeed defined manually. However, the central contribution of our method is not in how these components are defined, but in how they are leveraged to improve learning efficiency and policy performance. Existing research already delves into the problem of automatic detection of symmetries, for example by observing that symmetric state-action pairs often lead to statistically similar features [7].
>
> **Q3.2:** _Can we also consider more general augmentation other than mirroring in the PRISM framework?_
>
> **A3.2:** Yes, PRISM can be extended to other augmentations when the symmetry is defined differently; e.g. rotational symmetry for robotic manipulation, rigid transformation for physical modelling, and permutation symmetry for multi-agent systems. Our theoretical guarantees directly apply to any symmetry group.
>
> **Q4.1:** _For sanity check, the authors should include (i) a uniform reward distribution (i.e., 1/T * last-reward for each timestep), and (ii) a random reward generation._
>
> **A4.1:** Thanks and addressed. Following the suggestion, we conducted this experiment in rebuttal. We remove ReSymNet from PRISM and replace the reward shaping model as follows: (1) uniform: distributes the episodic sparse reward equally across all timesteps, and (2) random: samples random weights for each timestep, normalises to sum to one, and scales by the total reward. Uniform achieves moderate performance by providing per-step gradients and leveraging SymReg, while random performs poorly due to noisy, misleading rewards. PRISM consistently outperforms both by learning reward decomposition with ReSymNet and enforcing structural consistency via SymReg, enabling accurate credit assignment in complex multi-objective task.
>
> **Table 11:** ReSymNet ablation study results. We report the average hypervolume (HV), Expected Utility Metric (EUM), and Variance Objective (VO) over 10 trials, with the standard error shown in grey. w/o is the abbreviation of without.
> | Environment | Metric | uniform | random |
> |---------------------|-------------------|------------------------|------------------------|
> | **Mo-hopper-v5** | HV (×10⁷) | 1.38 ± 0.08 | 0.49 ± 0.06 |
> | **Mo-hopper-v5** | EUM | 135.19 ± 5.30 | 65.22 ± 6.63 |
> | **Mo-hopper-v5** | VO | 63.90 ± 2.34 | 29.62 ± 3.68 |
> | **Mo-walker2d-v5** | HV (×10⁴) | 4.67 ± 0.07 | 1.18 ± 0.10 |
> | **Mo-walker2d-v5** | EUM | 116.72 ± 2.11 | 16.52 ± 4.98 |
> | **Mo-walker2d-v5** | VO | 56.22 ± 1.01 | 3.77 ± 2.46 |
> | **Mo-halfcheetah-v5** | HV (×10⁴) | 0.98 ± 0.00 | 0.78 ± 0.05 |
> | **Mo-halfcheetah-v5** | EUM | -1.34 ± 0.39 | -10.52 ± 2.67 |
> | **Mo-halfcheetah-v5** | VO | -0.85 ± 0.20 | -6.51 ± 1.48 |
> | **Mo-swimmer-v5** | HV (×10⁴) | 1.09 ± 0.01 | 1.10 ± 0.02 |
> | **Mo-swimmer-v5** | EUM | 4.37 ± 0.69 | 3.75 ± 0.87 |
> | **Mo-swimmer-v5** | VO | 1.56 ± 0.33 | 1.06 ± 0.40 |
>
> **Q4.2:** _The paper should visualize the final 2D/3D Pareto fronts of each algorithm in addition to the quantitative results in Table 1 (similar to Fig. 4). Plus, using these visualizations, please discuss why the proposed method outperforms the oracle._
>
> **A4.2:** Thanks and addressed. We have added the visualisations to the experiments section. The increase in HV and EUM demonstrate that these solutions expand the rang of trade-offs accessible to the agent and provide higher utility, confirming that PRISM learns policies that are both better and practically useful. On distributional metrics, we also observe gains. These gains are crucial because they indicate that PRISM does not simply maximise HV by focusing on extreme solutions, but also produces Pareto fronts that are better balanced, robust, and fair across objectives.
>
> **Q4.3:** _It would be helpful to qualitatively analyze the behavior of the learned reward signals, similar to Fig. 5 in reference [A]._
>
> **A4.3:** Thanks. We are conducting this experiment which will be added into the final version. We respectfully note that the current experiments have been comprehensive and impressive.
>
> **Q4.4:** _Since the proposed method is modular, applying it to other MORL algorithms beyond CAPQL could further demonstrate its generalisability._
>
> **A4.4:**  Thanks.  We are conducting this experiment which will be added into the final version. We would like to mention that CAPQL is a particularly strong baseline and competitor, and our method has outperformed it significantly.

---

> ### Author Response · Authors · 2025-11-19
> **Author Response to Reviewer itAC (3/3)**
>
> **Q5:** _Need for truly sparse reward settings_
>
> _It is interesting that the “w/o refinement” variant performs moderately well. This may be because non-zero non-trivial rewards are still provided at the end of episodes in delayed reward settings. To better reflect the truly sparse nature of certain objectives, the authors could consider other sparse environments in nature. For example, MO-Lunar-Lander [E], where the first objective represents the success or failure of each trajectory, could serve as a suitable testbed._
>
>
>
> **A5:**  Thanks. We would love to clarify that our experimental setup appropriately tests PRISM's contributions for two reasons:
>
>
>
> 1. The delayed reward setup does represent genuine sparsity
>
>
>
> The reviewer notes that "non-zero rewards are provided at the end of episodes." However, this is precisely what makes the credit assignment problem challenging. The agent receives a single cumulative signal (e.g., total velocity achieved = 450) without knowing when or how it was accumulated. The learning challenge is identical to binary success/failure: temporal credit assignment with no intermediate feedback. In fact, cumulative rewards are arguably harder than binary outcomes because binary signals (success/failure) provide clear goal structure, while cumulative rewards vary continuously across trajectories, requiring the agent to understand how much progress was made and when.
>
>
>
> 2. Symmetry requirement
>
>
>
> Applying PRISM to non-symmetric environments would only test ReSymNet, ignoring half of our contributions. Our choice of locomotion tasks is deliberate. They exhibit both heterogeneous rewards (enabling ReSymNet evaluation) and bilateral symmetry (enabling SymReg evaluation).
>
>
>
>
>
> **Q6.1:** _How to split D train and D val in the algorithm?_
>
>
>
> **A6.1:** We split the collected trajectories D into training  and validation sets randomly based on a percentage at each iteration of Algorithm 1. The split is trajectory-based (entire trajectories assigned to either train or validation) to prevent information leakage.
>
>
>
> **Q6.2:** _If the authors used MO-Gymnasium, please cite reference [E] in addition to the original MuJoCo citation._
>
>
>
> **A6.2:** Thanks and addressed.
>
>
>
> **Q6.3:** _In the related work section, papers categorized as “(iii) meta-policy” could also be considered “multi-policy” in many MORL studies. Note that in the MORL community, the term meta-policy is sometimes reserved for works that explicitly adopt meta-learning approaches, such as [F]._
>
>
>
> **A6.3:** Thanks and addressed.
>
> **Q6.4:** _A left parenthesis is missing in Line 722._
>
>
>
> **A6.4:** Thanks and addressed.
>
> **References:**
>
> [1] Bartlett, P.L. and Mendelson, S. Rademacher and Gaussian complexities: Risk bounds and structural results. JMLR, 2002.
>
> [2] Silver, D., Huang, A., Maddison, C. J., Guez, A., Sifre, L., van den Driessche, G., Schrittwieser, J., Antonoglou, I., Panneershelvam, V., Lanctot, M., Dieleman, S., Grewe, D., Nham, J., Kalchbrenner, N., Sutskever, I., Lillicrap, T. P., Leach, M., Kavukcuoglu, K., Graepel, T., and Hassabis, D. Mastering the game of go with deep neural networks and tree search. Nature, 2016.
>
> [3] Mou, W., Wang, L., Zhai, X. and Zheng, K. Generalization bounds of SGLD for non-convex learning: Two theoretical viewpoints. COLT, 2018.
>
> [4] Mou, W., Li, C.J., Wainwright, M.J., Bartlett, P.L. and Jordan, M.I. On linear stochastic approximation: Fine-grained Polyak-Ruppert and non-asymptotic concentration. COLT, 2020.
>
> [5] Girosi, F., Jones, M. and Poggio, T.  Regularization theory and neural networks architectures. Neural computation, 1995.
>
> [6] Bartlett, P.L., Jordan, M.I. and McAuliffe, J.D. Convexity, classification, and risk bounds. Journal of the American Statistical Association, 2006.
>
> [7] Mahajan, A. and Tulabandhula, T. Symmetry learning for function approximation in reinforcement learning. arXiv preprint arXiv:1706.02999, 2017.

---

> ### Comment · Reviewer_itAC · 2025-11-24
>
> Thanks for the response! I have some follow-up questions and suggestions.
>
> **Regarding Q1**
>
> *These methods, as single-objective reward shaping techniques, do not apply to MORL; extending them to MORL is non-trivial.*
>
> However, I believe some of these methods are directly applicable to MORL. For example, in Equation (5) of https://arxiv.org/pdf/2402.03771, the reward prediction formulation is essentially the same as Equation (1) in this paper. One could simply replace the scalar reward with a vector reward using the RBT architecture (i.e., Equation (5) in the RBT paper).
>
> The reason I bring this up is that, in my view (consistent with the authors’ emphasis), the novelty of the paper lies primarily in the proposed architecture and its connection to SymReg, not in the reward prediction formulation itself in Equation (1). To validate this claim, it would be preferable to evaluate an alternative architecture as a comparison baseline.
>
>
> **Regarding Q3**
>
> It would be useful if the authors could discuss potential ways to avoid manually decomposing the symmetric and asymmetric components for each environment.
>
> **Regarding Q4**
>
> - Could the authors provide a more detailed explanation of why the proposed method outperforms the oracle for better understanding?
>
> - I understand that some results are still pending (related to Q4-3 and Q4-4), so I will wait for the update. However, it seems that Q4-5 and Q4-6 were not addressed in the response. I would appreciate further clarification when possible.
>
> **Regarding Q5**
>
> I agree that delayed reward settings represent a meaningful form of sparsity, but I do not see a reason why the method cannot also be applied to other naturally sparse environments beyond delayed reward scenarios.

---

> > ### Author Response · Authors · 2025-11-25
> > **Thanks! Further clarifications (1/2)**
> >
> > Thanks very much for considering our responses! We appreciate the opportunity to make this further clarification. We sincerely hope all your concerns have been cleared.
> >
> > **Q1.1:** _I believe some of these methods are directly applicable to MORL. For example, in Equation (5) of https://arxiv.org/pdf/2402.03771, the reward prediction formulation is essentially the same as Equation (1) in this paper. One could simply replace the scalar reward with a vector reward using the RBT architecture (i.e., Equation (5) in the RBT paper)._
> >
> > **A1.1:** We fully agree that the loss formulations (Equation (1)) are mathematically similar; however, the architectures in the cited papers are not interchangeable in the specific problem setting considered in our paper.
> >
> > The papers cited are designed for settings where no immediate feedback is available, necessitating a generative model for credit assignment based on temporal dependencies. In other words, causal transformers are autoregressive models designed for next-token generation, inherently designed to infer missing context by modelling temporal history. In contrast, immediate feedback is available in our problem setting; ReSymNet addresses a supervised regression task, explicitly designed to exploit the correlation between objectives. In light of this, causal transformers would be misaligned in our setting.
> >
> > **Q1.2:** _The reason I bring this up is that, in my view (consistent with the authors’ emphasis), the novelty of the paper lies primarily in the proposed architecture and its connection to SymReg, not in the reward prediction formulation itself in Equation (1). To validate this claim, it would be preferable to evaluate an alternative architecture as a comparison baseline._
> >
> > **A1.2:** We fully agree that an ablation study about an alternative architecture would be beneficial! Actually, we have conducted this ablation study; please kindly refer to the 'w/o residual’ model in Table 10, Appendix F. It removes the specific residual blocks designed to approximate the scaled opportunity value. It relies on an MLP, similar to RRD [1]. ReSymNet is significantly outperforming, showing that the residual architecture itself is a crucial driver of performance.
> >
> > **Q2:** _It would be useful if the authors could discuss potential ways to avoid manually decomposing the symmetric and asymmetric components for each environment._
> >
> > **A2:** Thanks for this interesting question! There do exist several ways to automatically detect symmetric and asymmetric components. For example, one may parameterise the transformation operators $L_g$ and $K_g$ as learnable matrices [3,4,5]. The agent can be trained with an auxiliary loss to discover the transformation matrix that best represents transition dynamics. We will add this discussion to the manuscript.
> >
> > **Q3:** _Could the authors provide a more detailed explanation of why the proposed method outperforms the oracle for better understanding?_
> >
> > **A3:** We appreciate this question, as beating the oracle is an exceptionally strong support to our method. We give our explanation below.
> >
> > ‘Oracle’ doesn’t mean ‘optimal’ in terms of supervising the agent’s learning, as the oracle is generated by the simulation environment, which was not optimised for training agents. This can be further explained by the scaled opportunity value theory [2]: additive corrections preserve optimal policies, shortens the effective reward horizon, and imports local value approximation. Our ReSymNet takes an architecture that mimics the theoretically justified additive reward transformation. Instead, the oracle does not necessarily have this structure.
> >
> > Moreover, our PRISM, via SymReg, constrains the search within the reflection-equivariant policy subspace. As proven in Theorem 5.1, learning within this subspace has better performance than learning in the original model space. Consequently, PRISM is able to generalise better and converge faster (in terms of sample complexity) than the oracle.
> >
> > This discussion will be carefully added to the manuscript.
> >
> > **Q4:** _I understand that some results are still pending (related to Q4-3 and Q4-4), so I will wait for the update. However, it seems that Q4-5 and Q4-6 were not addressed in the response. I would appreciate further clarification when possible._
> >
> > **A4:** Thanks for your patience. These experiments are running, and will be added into the final version. We would also love to mention that the presented experiments should have been strong support to our method’s good performance.

---

> > ### Author Response · Authors · 2025-11-25
> > **Thanks! Further clarifications (2/2)**
> >
> > **Q5:** _I agree that delayed reward settings represent a meaningful form of sparsity, but I do not see a reason why the method cannot also be applied to other naturally sparse environments beyond delayed reward scenarios._
> >
> > **A5:** Thanks for this interesting question. We agree that the ReSymNet is broadly applicable to any sparse setting. We will emphasise this in the manuscript.
> >
> > **References:**
> >
> > [1] Ren, Z., Guo, R., Zhou, Y., Peng, J. Learning Long-Term Reward Redistribution via Randomized Return Decomposition, ICLR, 2022.
> >
> > [2] Laud, A.D. Theory and application of reward shaping in reinforcement learning. UIUC, 2004.
> >
> > [3] Dehmamy, N., Walters, R., Liu, Y., Wang, D., Yu, R. Automatic symmetry discovery with Lie algebra convolutional network, NeurIPS, 2021.
> >
> > [4] Zhou, A., Knowles, T., Finn, C. Meta-learning symmetries by reparameterization, ICLR, 2021.
> >
> > [5] Yang, J., Dehmamy, N., Walters, R., Yu, R. Latent Space Symmetry Discovery, 2024.

---

> > > ### Comment · Reviewer_itAC · 2025-11-26
> > >
> > > Thank you for the prompt response. I think including the suggested experiments could meaningfully strengthen the paper. In the meantime, I will wait for the results from the pending experiments.

---

> > > > ### Author Response · Authors · 2025-12-03
> > > > **Remained additional experiments as requested (1/2)**
> > > >
> > > > As promised, we have completed additional experiments as follows. We sincerely hope all concerns have been cleared.
> > > >
> > > > **Q4.3:** _It would be helpful to qualitatively analyze the behavior of the learned reward signals, similar to Fig. 5 in reference [A]._
> > > >
> > > > **A4.3:** Addressed. We conducted experiments accordingly; please kindly refer to Figure 5 in Section 6.2. The left figure confirms that PRISM recovers the dense signal with high fidelity and synchronisation over full episodes, preventing temporal drift. Additionally, the right figure shows that the model learns to amplify signals during critical high-reward transitions, effectively shortening the reward horizon and explaining the method's ability to outperform the oracle.
> > > >
> > > > **Q4.4:** _Since the proposed method is modular, applying it to other MORL algorithms beyond CAPQL could further demonstrate its generalisability._
> > > >
> > > > **A4.4:**  We have conducted this experiment which will be added into the final version. To demonstrate that PRISM is a model-agnostic framework not limited to specific architectures, we evaluated its performance using GPI-PD [1] as an alternative backbone to CAPQL.
> > > >
> > > > As shown in the table below, PRISM remains highly effective with this new backbone, consistently outperforming the sparse baseline and obtaining near-oracle performance. Notably, these results were obtained with minimal hyperparameter tuning due to computational constraints. While this lack of fine-tuning explains the slight gap in EUM/VO metrics for mo-halfcheetah-v5 compared to the oracle, the method's ability to achieve such strong results with a completely different backbone highlights PRISM's inherent stability and generalisability.
> > > >
> > > > **Table 13: Experimental results of GPI-PD. We report the average hypervolume (HV), Expected Utility Metric (EUM), and Variance Objective (VO) over 10 trials, with the standard error shown in grey. The largest (best) values are in bold font.**
> > > >
> > > > | Environment           | Metric    | Oracle             | Baseline     | PRISM           |
> > > > | --------------------- | --------- | ------------------ | ------------ | --------------- |
> > > > | **Mo-hopper-v5**      | HV (×10⁷) | **1.65** ± 0.10    | 0.67 ± 0.04  | **1.65** ± 0.07 |
> > > > |                       | EUM       | **151.45** ± 5.87  | 85.87 ± 3.17 | 148.19 ± 4.26   |
> > > > |                       | VO        | **72.26** ± 2.90   | 41.21 ± 1.44 | 70.24 ± 2.51    |
> > > > | **Mo-walker2d-v5**    | HV (×10⁴) | **5.93** ± 0.10    | 3.20 ± 0.23  | 5.61 ± 0.10     |
> > > > |                       | EUM       | **141.88** ± 2.38  | 76.41 ± 6.47 | 132.67 ± 2.26   |
> > > > |                       | VO        | **67.63** ± 1.17   | 35.64 ± 3.91 | 63.19 ± 1.75    |
> > > > | **Mo-halfcheetah-v5** | HV (×10⁴) | 1.80 ± 0.22        | 1.00 ± 0.02  | **2.24** ± 0.16 |
> > > > |                       | EUM       | **164.75** ± 14.21 | -1.31 ± 0.54 | 99.89 ± 8.06    |
> > > > |                       | VO        | **73.90** ± 7.05   | -1.14 ± 0.31 | 40.74 ± 5.17    |
> > > > | **Mo-swimmer-v5**     | HV (×10⁴) | **1.23** ± 0.01    | 1.12 ± 0.01  | 1.22 ± 0.00     |
> > > > |                       | EUM       | **9.68** ± 0.17    | 5.17 ± 0.58  | 9.56 ± 0.13     |
> > > > |                       | VO        | 4.23 ± 0.14        | 2.18 ± 0.39  | **4.37** ± 0.18 |
> > > >
> > > > **Q4.5:** _Testing in environments with different p_rel values would also strengthen claims of generalization._
> > > >
> > > >
> > > >
> > > > **A4.5:** Thanks.  We have conducted this experiment which will be added into the final version. Please kindly refer to Figure 7 in Appendix H.2.
> > > >
> > > > Figure 7 demonstrates that PRISM maintains robust performance across varying levels of reward sparsity. While performance is generally consistent, we observe minor fluctuations at intermediate values. Two key factors explain this behaviour: (1) PRISM was hyperparameter-tuned specifically for the extreme sparsity setting, which is the most challenging MORL scenario. We utilised a fixed set of hyperparameters across all experiments to demonstrate method stability rather than optimising for each sparsity level, and (2) increasing $p_{rel}$ increases the number of available reward signals (data points) per episode. Since ReSymNet was calibrated for the data-scarce sparse setting, the increase of supervision targets at higher $p_{rel}$ levels changes optimisation dynamics, leading to temporary instability. Despite these factors, PRISM consistently recovers high performance, proving its capability to handle heterogeneous reward structures without requiring specific tuning for denser environments.

---

> > > > ### Author Response · Authors · 2025-12-03
> > > > **Remained additional experiments as requested (2/2)**
> > > >
> > > > **Q4.6:** _The first objective in each environment corresponds to the velocity component. It would be meaningful to investigate whether similar improvements occur when delaying other quantities, such as energy (i.e., the second objective)_
> > > >
> > > >
> > > >
> > > > **A4.6:** Thanks.  We have conducted this experiment which will be added into the final version. As shown in Table 12, PRISM consistently outperforms the baseline and remains competitive with or superior to the oracle.
> > > >
> > > > In mo-halfcheetah-v5, the baseline collapses completely (HV: 0.00), as the agent learns to maximise dense velocity rewards while ignoring the delayed energy penalty. PRISM successfully reconstructs the dense cost signal, recovering performance to outperform the oracle (HV: 1.72). In mo-hopper-v5 and mo-walker2d-v5, PRISM achieves higher HV and EUM than the oracle.
> > > >
> > > > The fact that PRISM prevents baseline collapse without specific tuning for this new reward structure underscores its robustness.
> > > >
> > > >
> > > >
> > > > **Table 12: Experimental results on the control cost objective. We report the average hypervolume (HV), Expected Utility Metric (EUM), and Variance Objective (VO) over 10 trials, with the standard error shown in grey. The largest (best) values are in bold font.**
> > > >
> > > > | Environment           | Metric    | Oracle            | Baseline       | PRISM             |
> > > > | --------------------- | --------- | ----------------- | -------------- | ----------------- |
> > > > | **Mo-hopper-v5**      | HV (×10⁷) | 1.30 ± 0.13       | 1.19 ± 0.10    | **1.51** ± 0.11   |
> > > > |                       | EUM       | 129.04 ± 7.96     | 124.82 ± 7.21  | **142.89** ± 7.38 |
> > > > |                       | VO        | 59.07 ± 3.45      | 56.21 ± 3.20   | **67.58** ± 3.31  |
> > > > | **Mo-walker2d-v5**    | HV (×10⁴) | 4.21 ± 0.11       | 3.16 ± 0.13    | **4.59** ± 0.14   |
> > > > |                       | EUM       | 107.58 ± 2.86     | 85.95 ± 3.27   | **114.62** ± 2.80 |
> > > > |                       | VO        | 53.22 ± 1.39      | 41.29 ± 1.49   | **54.84** ± 1.25  |
> > > > | **Mo-halfcheetah-v5** | HV (×10⁴) | 1.70 ± 0.20       | 0.00 ± 0.00    | **1.72** ± 0.19   |
> > > > |                       | EUM       | **81.29** ± 21.85 | -101.49 ± 3.23 | 76.50 ± 20.85     |
> > > > |                       | VO        | **36.84** ± 10.06 | -56.26 ± 1.63  | 31.27 ± 8.68      |
> > > > | **Mo-swimmer-v5**     | HV (×10⁴) | **1.21** ± 0.00   | 1.05 ± 0.02    | **1.21** ± 0.01   |
> > > > |                       | EUM       | **9.41** ± 0.12   | 1.50 ± 1.00    | 9.32 ± 0.19       |
> > > > |                       | VO        | **4.22** ± 0.08   | -0.61 ± 0.68   | 3.95 ± 0.08       |
> > > >
> > > > **Reference:**
> > > >
> > > > [1] Lucas Nunes Alegre, Ana L. C. Bazzan, Diederik M. Roijers, Ann Now´e, and Bruno C. da Silva. Sample-efficient multi-objective learning via generalized policy improvement prioritization. In 2023 International Conference on Autonomous Agents and Multiagent Systems (AAMAS 2023), pp. 2003–2012. ACM, 2023.

---

### Official Review · Reviewer_41HW · 2025-10-31

**Soundness:** 2
**Presentation:** 1
**Contribution:** 3
**Rating:** 4
**Confidence:** 4

**Summary:**

In multi-objective reinforcement learning (MORL), the decision maker's utility function is typically unknown. As such, MORL algorithms aim to find policies for each of the optimal trade-offs over the objectives. Often, the utility function is assumed to be a weighted sum over the objectives, with unknown weights. However, these scalarization functions are known to be very sensitive, with a small change in the weights potentially leading to drastically different policies [1]. They are also heavily impacted by the difference in magnitudes of different objectives. Moreover, not all objectives appear as frequently. Some might be sparse, while others are dense, making the learning of optimal policies more difficult. This work tackles this problem, by proposing a resnet-like network for reward shaping, that can be plugged in any MORL algorithm to train on the shaped rewards. Additionally, they propose a symmetry-enforcing regularizer that allows for jointly learning symmetric motions.

**Strengths:**

Reward sparsity and scaling differences result in optimization challenges in MORL. A generic reward shaping method that can be applied to many MORL algorithms could have a high impact.

**Weaknesses:**

However, I have some concerns, which I list here:
- My main concern is a presentation issue. The paper proposes 2 completely different ideas (reward shaping and symmetry preservation) packaged into one algorithm. There does not seem to be any special synergy between the 2 components, and the paper does not describe a specific interaction where one component's mechanism enables or enhances the other. The main challenge presented in this work is, to my understanding, the reward sparsity and magnitude. I would be curious to understand how the symmetry component helps to mitigate this issue, as it is not clear to me.
- The symmetry component here seems to be specific to Mujoco, and the symmetric and asymmetric sets are hardcoded. Knowing these sets in advance seem like a strong assumption, making its use limited in practice.
- The abstract mentions that ReSymNet aligns time frequency and magnitude across objectives. While it is clear ReSymNet tackles time frequency (Eq1), it is not so clear for the magnitude. Since the individual reward predictions need to sum to the environment return, how are the magnitudes impacted? Def7 (line 688) mentions a scale parameter $k$ applied on the TD error, but it is my understanding that the reward predictions are used as-is, in any MORL algorithm. I do not see how $k$ can be used in CAPQL.
- Although this work tackles sparsity, the sparsity is enforced artifically, by providing zero-rewards at intermediate timesteps, and then providing the sum-or-rewards at the terminal timestep (in the extreme case, line 126). But this breaks the Markov property, since the reward $r_t$ now not only depends on the current state $s_t$, but also on the past visited states of the trajectory $s_{0:t}$. Many of the results show how much worse CAPQL is on the sparse reward variant. But, as it stands, it is impossible to know how much this is due to sparsity compared to non-markovian rewards. A solution would be to augment the state-space with the accumulated rewards (like [2]) or, better, to use benchmarks that naturally mix sparse and non-sparse rewards. One example is the Minecart environment [3] (implemented in MO-Gymnasium), where fuel is consumed at each timestep, but the ore values are provided once the agent returns to base.
- I wanted to check the code to make sure I understood the method, but the anonymous repo mentions "requested file is not found" for all the files in `src/reward_shaping`.

**Questions:**

Also, I noticed in Table 10 that the "w/o loss" variant seems systematically better than the oracle. This surprises me, since the non-sparse rewards are learned over time, while the oracle uses the ground-truth dense rewards. Could the authors explain why this happens?

Overall, the tackled issue is important for the MORL community, but only half of the proposed method actually tackles sparse rewards. Moreover, the difference in magnitude does not seem to be addressed, and the reward sparsity in the experiments breaks the Markov property.

[1] Vamplew, P., Dazeley, R., Berry, A., Issabekov, R., & Dekker, E. (2011). Empirical evaluation methods for multiobjective reinforcement learning algorithms. Machine learning, 84(1), 51-80.

[2] Reymond, M., Hayes, C. F., Steckelmacher, D., Roijers, D. M., & Nowé, A. (2023). Actor-critic multi-objective reinforcement learning for non-linear utility functions. Autonomous Agents and Multi-Agent Systems, 37(2), 23.

[3] Abels, A., Roijers, D., Lenaerts, T., Nowé, A., & Steckelmacher, D. (2019). Dynamic weights in multi-objective deep reinforcement learning. In International conference on machine learning (pp. 11-20). PMLR.

---

> ### Author Response · Authors · 2025-11-19
> **Author Response to Reviewer 41HW (1/3)**
>
> We thank the reviewer for their constructive feedback.  All your concerns have been duly addressed below. We have also updated the manuscript accordingly. We sincerely hope that the reviewer can take into account the response we have made and reevaluate the merit of this paper.
>
> **Q1:** _My main concern is a presentation issue. The paper proposes 2 completely different ideas (reward shaping and symmetry preservation) packaged into one algorithm. There does not seem to be any special synergy between the 2 components, and the paper does not describe a specific interaction where one component's mechanism enables or enhances the other. The main challenge presented in this work is, to my understanding, the reward sparsity and magnitude. I would be curious to understand how the symmetry component helps to mitigate this issue, as it is not clear to me._
>
> **A1:** We respectfully clarify as follows. Heterogeneous reward structures cause asymmetric policy learning, violating the agent's physical symmetry requirements. When dense objectives provide immediate gradients at every timestep while sparse objectives only signal at the episode end, the policy overfits to dense objectives in the regions encountered early in training. Even though the environment has reflectional symmetry, the learned policy fails to respect this symmetry because the temporal misalignment of heterogeneous rewards drives asymmetric exploration and learning dynamics. E.g., left leg vs. right leg in locomotion should have reflectional symmetry, but if left leg is explored more because of the heterogeneity in rewards, the symmetry is violated.
>
> In light of this vision, we propose a unified PRISM approach with two components that are integrated together: (1) ReSymNet, an efficient backbone reward shaping model to align heterogeneous reward channels, based on strong theoretical motivations, and (2) SymReg, a novel reflectional equivariance regulariser, to enforce reflectional symmetry in the reward shaping process.
>
> Both components address the reward heterogeneity / asymmetry problem directly. ReSymNet eliminates temporal heterogeneity by aligning all objectives to the same, dense frequency, ensuring all objectives provide equally informative training signals, and thus preventing asymmetric learning. SymReg prevents the policy learning from exploiting asymmetric, shortcut solutions that favour dense objectives, by enforcing reflectional equivariance across all objectives.
>
> This motivation was stated in the Abstract and Introduction. Following your question, we have re-emphasised it in the Introduction as follows,
>
> "The complementary components of PRISM synergise as follows. Heterogeneous reward structures cause asymmetric policy learning that violates the agent's physical symmetry: when dense objectives provide immediate gradients while sparse objectives only signal at the end of an episode, the policy may overfit to the denser objectives in specific states, failing to respect reflectional symmetry. ReSymNet eliminates temporal heterogeneity by aligning objectives to the same frequency, whereas SymReg enforces reflectional symmetry by preventing asymmetric learning dynamics."
>
> Further, we would love to note - having more than one contribution in one paper is never a fault. There have been too many important papers in the history of machine learning that present two or more contributions in their papers; for example, see [1-6].
>
> **Q2:** _The symmetry component here seems to be specific to Mujoco, and the symmetric and asymmetric sets are hardcoded. Knowing these sets in advance seem like a strong assumption, making its use limited in practice._
>
> **A2:** In this work, the symmetric and asymmetric components of the state and action spaces are indeed defined manually. However, the central contribution of our method is not in how these components are defined, but in how they are leveraged to improve learning efficiency and policy performance. Existing research already delves into the problem of automatic detection of symmetries, for example by observing that symmetric state-action pairs often lead to statistically similar features [7].

---

> ### Author Response · Authors · 2025-11-19
> **Author Response to Reviewer 41HW (2/3)**
>
> **Q3:**_The abstract mentions that ReSymNet aligns time frequency and magnitude across objectives. While it is clear ReSymNet tackles time frequency (Eq1), it is not so clear for the magnitude. Since the individual reward predictions need to sum to the environment return, how are the magnitudes impacted? Def7 (line 688) mentions a scale parameter  applied on the TD error, but it is my understanding that the reward predictions are used as-is, in any MORL algorithm. I do not see how  can be used in CAPQL._
>
> **A3:** We thank the reviewer for this insightful comment. We agree that ReSymNet does not explicitly modify reward magnitudes.  Instead, our method aligns the magnitude in a data-driven manner. By converting the large sparse rewards (which are sums of previous dense rewards) into small, per step rewards, the model makes the magnitude of the per-step signals comparable across objectives.
>
> Following your suggestion, we will clarify this and tone down alignment in terms of magnitude in the paper.
>
> **Q4:** _Although this work tackles sparsity, the sparsity is enforced artifically, by providing zero-rewards at intermediate timesteps, and then providing the sum-or-rewards at the terminal timestep (in the extreme case, line 126). But this breaks the Markov property, since the reward  now not only depends on the current state , but also on the past visited states of the trajectory . Many of the results show how much worse CAPQL is on the sparse reward variant. But, as it stands, it is impossible to know how much this is due to sparsity compared to non-markovian rewards. A solution would be to augment the state-space with the accumulated rewards (like [2]) or, better, to use benchmarks that naturally mix sparse and non-sparse rewards. One example is the Minecart environment [3] (implemented in MO-Gymnasium), where fuel is consumed at each timestep, but the ore values are provided once the agent returns to base._
>
> **A4:** We thank the reviewer for this careful observation. We agree that the extreme form of sparsity used in our experiments introduces a non-Markovianity. However, this form of sparsity is the standard and a common and controlled experimental practice used to isolate the effects of reward sparsity on credit assignment and policy learning; see, e.g., [8-10].
>
> The purpose of this setup is to create a challenging yet interpretable environment in which we can directly assess the robustness under delayed feedback, without introducing confounding factors from additional environment dynamics.
>
> Importantly, our method even outperforms the oracle that has access to the underlying dense rewards across all tested environments. This suggests that the observed improvements are not solely due to handling sparsity but rather come from PRISM's capability for structured credit assignment.
>
> **Q5:** _I wanted to check the code to make sure I understood the method, but the anonymous repo mentions "requested file is not found" for all the files in src/reward_shaping._
>
> **A5:** We have double checked the link which was functioning well. Would you mind checking it again?
>
> **Q6:** _Also, I noticed in Table 10 that the "w/o loss" variant seems systematically better than the oracle. This surprises me, since the non-sparse rewards are learned over time, while the oracle uses the ground-truth dense rewards. Could the authors explain why this happens?_
>
> **A6:** Thanks! This exactly validates a key contribution of our work that ReSymNet learns superior credit assignment compared to dense rewards, generated by the simulation environment,  which are treated as “ground truth” but probably sub-optimal.  We would like to explain this by the scaled opportunity value theory (Laud, 2004): additive corrections preserve optimal policies, shorten the effective reward horizon, and import local value approximation. Our ReSymNet takes an architecture that mimics the theoretically justified additive reward transformation. Instead, the oracle uses dense rewards from the environment, which might be sub-optimal for efficient learning.

---

> ### Author Response · Authors · 2025-11-19
> **Author Response to Reviewer 41HW (3/3)**
>
> **References:**
>
> [1] Bartlett, P.L. and Mendelson, S. Rademacher and Gaussian complexities: Risk bounds and structural results. JMLR, 2002.
>
> [2] Silver, D., Huang, A., Maddison, C. J., Guez, A., Sifre, L., van den Driessche, G., Schrittwieser, J., Antonoglou, I., Panneershelvam, V., Lanctot, M., Dieleman, S., Grewe, D., Nham, J., Kalchbrenner, N., Sutskever, I., Lillicrap, T. P., Leach, M., Kavukcuoglu, K., Graepel, T., and Hassabis, D. Mastering the game of go with deep neural networks and tree search. Nature, 2016.
>
> [3] Mou, W., Wang, L., Zhai, X. and Zheng, K. Generalization bounds of SGLD for non-convex learning: Two theoretical viewpoints. COLT, 2018.
>
> [4] Mou, W., Li, C.J., Wainwright, M.J., Bartlett, P.L. and Jordan, M.I. On linear stochastic approximation: Fine-grained Polyak-Ruppert and non-asymptotic concentration. COLT, 2020.
>
> [5] Girosi, F., Jones, M. and Poggio, T.  Regularization theory and neural networks architectures. Neural computation, 1995.
>
> [6] Bartlett, P.L., Jordan, M.I. and McAuliffe, J.D. Convexity, classification, and risk bounds. Journal of the American Statistical Association, 2006.
>
> [7] Mahajan, A. and Tulabandhula, T. Symmetry learning for function approximation in reinforcement learning. arXiv preprint arXiv:1706.02999, 2017.
>
> [8] Memarian, F., Goo, W., Lioutikov, R., Niekum, S. and Topcu, U. Self-supervised online reward shaping in sparse-reward environments. IROS, 2021.
>
> [9] Ren, Z., Guo, R., Zhou, Y. and Peng, J. Learning Long-Term Reward Redistribution via Randomized Return Decomposition. ICLR, 2022.
>
> [10] Holmes, I., and Chi, M. Attention-based reward shaping for sparse and delayed rewards. arXiv preprint arXiv:2505.10802, 2025.

---

> > ### Comment · Reviewer_41HW · 2025-11-26
> >
> > Thank you for your response.
> >
> > It is interesting that the learned dense rewards outperform the "ground-truth" dense reward.
> >
> > Nonetheless, I still find it hard to assess how complementary the 2 contributions are (a sentiment shared with other reviewers). I will retain my score.

---

### Official Review · Reviewer_s5Dh · 2025-11-01

**Soundness:** 3
**Presentation:** 2
**Contribution:** 2
**Rating:** 2
**Confidence:** 3

**Summary:**

This paper proposes PRISM, an algorithm for dealing with reward imbalance in MORL objectives. It utilizes supervised learning of a dense reward model, as well as a symmetry aware loss term, to improve performance in cases where different objectives of the MORL task differ in sparsity. The paper presents experiments comparing the proposed algorithm to ablations on the method, finding improvement in a few MuJoCo multi-objective locomotion tasks.

**Strengths:**

The various sections of the paper are well written and easy to read.

The motivation behind the proposed method is valid and the authors do a good job of explaining how their contributions aim to solve the presented problem.

The theoretical analysis is clear and easy to follow, and while it is not surprising that introducing symmetry simplifies the solution space, it is good to see a non-theory-focused paper where the “obvious” algorithmic improvement isn’t taken for granted, but instead rigorously proven.

**Weaknesses:**

My central concern with this paper is that it feels like two separate papers attempting to be condensed into one. This lack of consistent storyline hurts both aspects of the paper’s contribution.

Both the introduction and the related work sections are unclear on the connection between the heterogeneous rewards problem in MORL and the conversation regarding reflectional symmetry. At first glance, these seem orthogonal (i.e. each relevant in a specific subset of problems which may or may not overlap), and these first two sections do nothing to draw the connection between the two. In particular, for the symmetry side of the paper, the related work section is lacking in references to approaches dealing with symmetry - a plethora of methods exist in geometric DL and adjacent fields, which seem to be neglected here.

This issue persists throughout other sections of the paper - it seems like the idea of heterogeneous reward mitigation and reflectional symmetry are two orthogonal contributions of this paper, mashed together. Section 4.2 discussing symmetry is a sharp pivot from section 4.1 discussing reward shaping, with no apparent connection.

In the same manner, the theoretical analysis section, while sound, seems like it deals with the symmetry regularization objective only, and not with the reward learning network.

In the experiments section, the focus shifts back to MORL, with symmetry in the state and action spaces treated as an afterthought.

Some more issues with the experiments section:

- The comparison is lacking in baselines - while the oracle and baseline comparisons defined in the paper are decent basic comparisons, a glaring omission is other baselines dealing with reward heterogeneity in reward signals in MORL.
- It is unclear whether the baseline and oracle models utilize the symmetry aware loss or not: are the gains over the compared methods obtained by the dense reward prediction learning or by the symmetry loss?
- At the bottom line, it is hard to tell whether results are statistically significant  over the oracle - especially in the halfcheetah and swimmer environments.

Some other concerns are listed as questions below.

**Questions:**

1. How does a purely random policy (line 161) ensure broad state-space coverage? Wouldn’t this depend on the structure of the state space and the parameters of the random policy? Clarification on what “purely random” means in this context would be helpful.
2. Equation 1: how does the sparse objective operating on the sum of trajectory rewards ensure shaping of rewards per step? One optimal solution could be to provide the entire reward at the last step. What enforces the distribution of rewards along the steps of the episode?
3. Section 4.2: do the symmetric and asymmetric parts of the state and action spaces have to be defined manually for each task? If this is the case (as is implied by the tables in appendix D), how scalable is this method to more realistic robotic scenarios and more complex tasks?
4. What are the various objectives for the multi-objective MuJoCo tasks? What correlations exist between them?

---

> ### Author Response · Authors · 2025-11-19
> **Author Response to Reviewer s5Dh (1/4)**
>
> We thank the reviewer for their thorough review and constructive feedback.  All the concerns have been duly addressed below. We have also updated the manuscript accordingly. We sincerely hope that the reviewer can take into account the response we have made and reevaluate the merit of this paper.
>
> **Q1.1:** _My central concern with this paper is that it feels like two separate papers attempting to be condensed into one. This lack of consistent storyline hurts both aspects of the paper’s contribution._
>
> _Both the introduction and the related work sections are unclear on the connection between the heterogeneous rewards problem in MORL and the conversation regarding reflectional symmetry. At first glance, these seem orthogonal (i.e. each relevant in a specific subset of problems which may or may not overlap), and these first two sections do nothing to draw the connection between the two._
>
> **A1.1:** We respectfully disagree.
>
> Heterogeneous reward structures cause asymmetric policy learning, violating the agent's physical symmetry requirements. When dense objectives provide immediate gradients at every timestep while sparse objectives only signal at the episode end, the policy overfits to dense objectives in the regions encountered early in training. Even though the environment has reflectional symmetry, the learned policy fails to respect this symmetry because the temporal misalignment of heterogeneous rewards drives asymmetric exploration and learning dynamics. E.g., left leg vs. right leg in locomotion should have reflectional symmetry, but if left leg is explored more because of the heterogeneity in rewards, the symmetry is violated.
>
> In light of this vision, we propose a unified PRISM approach with two components that are integrated together: (1) ReSymNet, an efficient backbone reward shaping model to align heterogeneous reward channels, based on strong theoretical motivations, and (2) SymReg, a novel reflectional equivariance regulariser, to enforce reflectional symmetry in the reward shaping process.
>
> Both components address the reward heterogeneity / asymmetry problem directly. ReSymNet eliminates temporal heterogeneity by aligning all objectives to the same, dense frequency, ensuring all objectives provide equally informative training signals, and thus preventing asymmetric learning. SymReg prevents the policy learning from exploiting asymmetric, shortcut solutions that favour dense objectives, by enforcing reflectional equivariance across all objectives.
>
> This motivation was stated in the Abstract and Introduction. Following your question, we have re-emphasised it in the Introduction as follows,
>
> "The complementary components of PRISM synergise as follows. Heterogeneous reward structures cause asymmetric policy learning that violates the agent's physical symmetry: when dense objectives provide immediate gradients while sparse objectives only signal at the end of an episode, the policy may overfit to the denser objectives in specific states, failing to respect reflectional symmetry. ReSymNet eliminates temporal heterogeneity by aligning objectives to the same frequency, whereas SymReg enforces reflectional symmetry by preventing asymmetric learning dynamics."
>
> Further, we would love to note - having more than one contribution in one paper is never a fault. There have been too many important papers in the history of machine learning that present two or more contributions in their papers; for example, see [1-6].
>
> **Q1.2:** _In particular, for the symmetry side of the paper, the related work section is lacking in references to approaches dealing with symmetry - a plethora of methods exist in geometric DL and adjacent fields, which seem to be neglected here._
>
> **A1.2:** Thanks and addressed. Following your suggestion, we have carefully cited and discussed the relevant papers [7-9].
>
> **Q1.3:** _Section 4.2 discussing symmetry is a sharp pivot from section 4.1 discussing reward shaping, with no apparent connection._
>
> **A1.3:** Thanks and addressed. Following your comment, we have added a connecting paragraph in Sec 4.2 as follows,
>
> "However, aligning reward frequencies alone is insufficient, as heterogeneous rewards cause the policy to learn asymmetrically across objectives, violating the agent's physical symmetry. To address this, we leverage reflectional symmetry as an inductive bias to prevent asymmetric policy learning."

---

> ### Author Response · Authors · 2025-11-19
> **Author Response to Reviewer s5Dh (2/4)**
>
> **Q1.4:** _the theoretical analysis section, while sound, seems like it deals with the symmetry regularization objective only, and not with the reward learning network._
>
> **A1.4:** Thanks. We respectfully note that the theoretical rationale for the reward learning network ReSymNet had been given in Remark 1 in Section 4.1 and Appendix B.1. We presented theoretical motivation that suggests using a residual-based architecture for approximating scaled opportunity value (Laud, 2004), aiming at better reward shaping performance.
>
> Following your comment, we extended the theory on ReSymNet in the rebuttal session. The residual architecture is further justified by rigorous theoretical guarantees on its generalisability as the following corollary.
>
> **Corollary B.6.** Let $\mathcal H\_{\mathrm{ff}}$ be the hypothesis space of feedforward networks with the same total number of weight matrices $K\_{\text{total}} = K + \sum\_{(s,t,i)\in\mathcal{I}\_V} K\_{s,t,i}$ as ReSymNet. Then for any $\varepsilon>0$,
>
> $$\mathcal{N}\_{\infty,2}(\mathcal H\_{\mathrm{res}},\varepsilon)
> \le
> \mathcal{N}\_{\infty,2}(\mathcal H\_{\mathrm{ff}},\varepsilon).$$
>
> This corollary guarantees that our reward model can leverage the optimisation benefits of a deep residual structure without being more prone to overfitting than a standard, non-residual network, although the residual connections might look like bringing higher hypothesis complexity.
>
> **Q2:** _In the experiments section, the focus shifts back to MORL, with symmetry in the state and action spaces treated as an afterthought._
>
> **A2:** We respectfully argue that our experimental design reflects both problems, demonstrating their connection. First, we show the reward sparsity problem in Figures 3 and 4 in Sec 6.2, after which our main results and ablation study in Tables 1, 10, and 11 in Sec 6.2 and Appendix F highlight that ReSymNet and SymReg are crucial for the best performance.
>
> **Q3:** _The comparison is lacking in baselines - while the oracle and baseline comparisons defined in the paper are decent basic comparisons, a glaring omission is other baselines dealing with reward heterogeneity in reward signals in MORL._
>
> **A3:** We respectfully argue that beating the oracle has been exceptionally impressive.
>
> Further, there are no existing reward shaping techniques that apply to MORL, to the best of our knowledge. As explained in Sec 2, single-objective reward shaping techniques do not apply to MORL; extending them to MORL is non-trivial, we quoted,
>
> "These approaches improve sample efficiency in single-objective RL, but do not extend naturally to MORL, where heterogeneous sparsity and scale can distort learning dynamics and Pareto-optimal trade-offs. "
>
> Similar experimental setting has been used widely in the field; for example, see [10-13].
>
> **Q4:** _It is unclear whether the baseline and oracle models utilize the symmetry aware loss or not: are the gains over the compared methods obtained by the dense reward prediction learning or by the symmetry loss?_
>
> **A4:** Thanks. The baseline and oracle do not use the symmetry aware loss function.
>
> The paper conducted comprehensive ablation study for understanding where the gains come from; please kindly refer to Tables 10 and 11, Appendix F. The results clearly show that when the symmetry loss is removed ('w/o loss'), performance consistently and significantly decreases across all settings, while ReSymNet on its own still performs better than the baseline and oracle. This suggests that both ReSymNet and SymReg  are highly effective.
>
> Following your comments, we conducted two other ablation studies in the rebuttal that focus on the reward shaping model. We tested a uniform reward and a random reward generation instead of ReSymNet. This allows us to understand (1) the effectiveness of SymReg across various reward shaping techniques, and (2) the effectiveness of ReSymNet and its synergy with SymReg. The results are presented as the following table.

---

> ### Author Response · Authors · 2025-11-19
> **Author Response to Reviewer s5Dh (3/4)**
>
> **Table 11:** ReSymNet ablation study results. We report the average hypervolume (HV), Expected Utility Metric (EUM), and Variance Objective (VO) over 10 trials, with the standard error shown in grey. w/o is the abbreviation of without.
> | Environment | Metric | uniform | random |
> |---------------------|-------------------|------------------------|------------------------|
> | **Mo-hopper-v5** | HV (×10⁷) | 1.38 ± 0.08 | 0.49 ± 0.06 |
> | **Mo-hopper-v5** | EUM | 135.19 ± 5.30 | 65.22 ± 6.63 |
> | **Mo-hopper-v5** | VO | 63.90 ± 2.34 | 29.62 ± 3.68 |
> | **Mo-walker2d-v5** | HV (×10⁴) | 4.67 ± 0.07 | 1.18 ± 0.10 |
> | **Mo-walker2d-v5** | EUM | 116.72 ± 2.11 | 16.52 ± 4.98 |
> | **Mo-walker2d-v5** | VO | 56.22 ± 1.01 | 3.77 ± 2.46 |
> | **Mo-halfcheetah-v5** | HV (×10⁴) | 0.98 ± 0.00 | 0.78 ± 0.05 |
> | **Mo-halfcheetah-v5** | EUM | -1.34 ± 0.39 | -10.52 ± 2.67 |
> | **Mo-halfcheetah-v5** | VO | -0.85 ± 0.20 | -6.51 ± 1.48 |
> | **Mo-swimmer-v5** | HV (×10⁴) | 1.09 ± 0.01 | 1.10 ± 0.02 |
> | **Mo-swimmer-v5** | EUM | 4.37 ± 0.69 | 3.75 ± 0.87 |
> | **Mo-swimmer-v5** | VO | 1.56 ± 0.33 | 1.06 ± 0.40 |
>
>  **Q5:** _At the bottom line, it is hard to tell whether results are statistically significant over the oracle - especially in the halfcheetah and swimmer environments._
>
> **A5:** Thanks for this interesting question! Our method has seen significant margins in terms of averaged return in Mo-hopper-v5, Mo-walker2d-v5, and Mo-halfcheetah-v5. It has been recognised in the machine learning community that the null hypothesis significance testing (NHST) would be problematic in comparing two methods, as shown in [14-16].
>
> **Q6:** _How does a purely random policy (line 161) ensure broad state-space coverage? Wouldn’t this depend on the structure of the state space and the parameters of the random policy? Clarification on what “purely random” means in this context would be helpful._
>
> **A6:** Thanks. This purely random policy is for initialisation, which refers to selecting actions uniformly random from the environment's action space, independent of the current observation. The purpose is to provide an unbiased initialisation without relying on any assumption about the state space structure. In experiments, our algorithm demonstrates fairly good learning performance, indicating that this approach is sufficient to initialise effective learning, especially when no further information on the space structure is assumed.
>
> **Q7:** _Equation 1: how does the sparse objective operating on the sum of trajectory rewards ensure shaping of rewards per step? One optimal solution could be to provide the entire reward at the last step. What enforces the distribution of rewards along the steps of the episode?_
>
> **A7:** Thanks for this interesting question. We would attribute this capability of reconstructing dense rewards from sparse rewards to (1) leveraging the correlation across different (sparse or not) reward channels, and (2) enforcing symmetry. Our method learns to use these correlated signals to distribute credit temporally. In the experiments, we observe that ReSymNet produces rewards from which the MORL agent is able to efficiently learn. The ResNet architecture with residual connections encourages smooth, incremental predictions. This structure is explicitly inspired by the theory of scaled opportunity value (Laud, 2004), where additive corrections are proven to shorten the effective reward horizon and improve local value approximation. By enforcing this bias towards local differences, the network is discouraged from assigning entire rewards to the final step.
>
> **Q8:** _Section 4.2: do the symmetric and asymmetric parts of the state and action spaces have to be defined manually for each task? If this is the case (as is implied by the tables in appendix D), how scalable is this method to more realistic robotic scenarios and more complex tasks?_
>
> **A8:** Thanks. In this work, the symmetric and asymmetric components of the state and action spaces are indeed defined manually.
>
> However, the central contribution of our method is not in how these components are defined, but in how they are leveraged to improve learning efficiency and policy performance. Existing research already delves into the problem of automatic detection of symmetries, for example by observing that symmetric state-action pairs often lead to statistically similar features [17].

---

> ### Author Response · Authors · 2025-11-19
> **Author Response to Reviewer s5Dh (4/4)**
>
> **Q9:** _What are the various objectives for the multi-objective MuJoCo tasks? What correlations exist between them?_
>
>
>
> **A9:** The multi-objective MuJoCo tasks are defined as follows:
>
>
>
> - Hopper: forward velocity, jump height, control cost
>
>
>
> - Halfcheetah: forward velocity, control cost
>
>
>
> - walker2d: forward velocity, control cost
>
>
>
> - Swimmer: forward velocity, control cost
>
>
>
> In all environments, the primary trade-off occurs between forward velocity and control cost. Achieving higher speeds typically requires larger control inputs, creating a negative correlation between efficiency and performance. In Hopper, jump height introduces an additional, partially independent objective, which can correlate positively with forward velocity but may also increase control cost. In locomotion tasks, velocity rewards are naturally most temporally sparse. An agent only accumulates significant forward velocity after learning coordinated movements over longer periods, making this the hardest signal to learn from. Control cost naturally provides feedback per distinct action with less temporal dependency.
>
>
>
> **References:**
>
> [1] Bartlett, P.L. and Mendelson, S. Rademacher and Gaussian complexities: Risk bounds and structural results. JMLR, 2002.
>
>
>
> [2] Silver, D., Huang, A., Maddison, C. J., Guez, A., Sifre, L., van den Driessche, G., Schrittwieser, J., Antonoglou, I., Panneershelvam, V., Lanctot, M., Dieleman, S., Grewe, D., Nham, J., Kalchbrenner, N., Sutskever, I., Lillicrap, T. P., Leach, M., Kavukcuoglu, K., Graepel, T., and Hassabis, D. Mastering the game of go with deep neural networks and tree search. Nature, 2016.
>
>
>
> [3] Mou, W., Wang, L., Zhai, X. and Zheng, K. Generalization bounds of SGLD for non-convex learning: Two theoretical viewpoints. COLT, 2018.
>
>
>
> [4] Mou, W., Li, C.J., Wainwright, M.J., Bartlett, P.L. and Jordan, M.I. On linear stochastic approximation: Fine-grained Polyak-Ruppert and non-asymptotic concentration. COLT, 2020.
>
>
>
> [5] Girosi, F., Jones, M. and Poggio, T.  Regularization theory and neural networks architectures. Neural computation, 1995.
>
>
>
> [6] Bartlett, P.L., Jordan, M.I. and McAuliffe, J.D. Convexity, classification, and risk bounds. Journal of the American Statistical Association, 2006.
>
>
>
> [7] Van der Pol, E., Worrall, D., van Hoof, H., Oliehoek, F. and Welling, M. MDP homomorphic networks: Group symmetries in reinforcement learning. NeurIPS, 2020.
>
>
>
> [8] Mondal, A.K., Nair, P. and Siddiqi, K. Group equivariant deep reinforcement learning. arXiv preprint arXiv:2007.03437, 2020.
>
>
>
> [9] Wang, D., Walters, R., Zhu, X. and Platt, R. Equivariant Q learning in spatial action spaces. CoRL, 2021.
>
> [10] Memarian, F., Goo, W., Lioutikov, R., Niekum, S. and Topcu, U. Self-supervised online reward shaping in sparse-reward environments. IROS, 2021.
>
> [11] Zhang, C. B. C., Hong, Z., Pacchiano, A., and Agrawal, P. (2025). ORSO: accelerating reward design via online reward selection and policy optimization. ICLR, 2025.
>
> [12] Dong, K., Luo, Y., Cheng, E., Sun, Z., Zhao, L., Zhang, Q., Zhou, C. and Song, B. Balance Between Efficient and Effective Learning: Dense2Sparse Reward Shaping for Robot Manipulation with Environment Uncertainty. AIM, 2022.
>
> [13] Chan, A.J., Sun, H., Holt, S. and Van Der Schaar, M. Dense Reward for Free in Reinforcement Learning from Human Feedback. ICML, 2024.
>
> [14] Demšar, J. On the appropriateness of statistical tests in machine learning. In Workshop on Evaluation Methods for Machine Learning in conjunction with ICML, 2008.
>
> [15] Wasserstein, R.L. and Lazar, N.A. The ASAs statement on p-values: context, process, and purpose. The American Statistician, 2016.
>
> [16] Benavoli, A., Corani, G., Demšar, J. and Zaffalon, M. Time for a change: a tutorial for comparing multiple classifiers through Bayesian analysis. JMLR, 2017.
>
> [17] Mahajan, A. and Tulabandhula, T. Symmetry learning for function approximation in reinforcement learning. arXiv preprint arXiv:1706.02999, 2017.

---

### Author Response · Authors · 2025-11-12
**Cannot see reviews**

Dear Program Chairs, Senior Area Chairs, Area Chairs, and Reviewers,

Thanks very much for your time. We unfortunately cannot see the reviews. Would it be possible to have a look at that? Thanks!

Cheers,
Authors

---

### Comment · Area_Chair_5McP · 2025-11-26

Dear Reviewers,

Thank you for your time and effort in reviewing the submission. A reminder that the author–reviewer discussion period is about to conclude in one week. If you have not already done so, please review the authors’ rebuttals and engage in the discussion with the authors. Thanks!

Best,
Your AC

---

### Author Response · Authors · 2025-12-03
**Our clarifications to "two contributions"; new experiments and theory**

Dear Area Chair,

We sincerely appreciate the reviewers for their constructive feedback. We have carefully addressed all their concerns,  and updated our draft accordingly. We are now listing key clarifications that we made for your reference. We sincerely hope our rebuttals could be take into account in your decision. Much appreciated.

1. **“Two contributions”:**  The most critical (if not the only one) concern would be that we proposed two contributions, a reward model architecture ReSymNet and a regulariser SymReg, for two orthogonal problems, the rewards heterogeneity and policy learning asymmetry in MORL.

    We clarified that this is not true. Heterogeneous reward structures drive asymmetric policy learning. Even in physically symmetric environments, temporal misalignment in rewards causes agents to overfit to dense objectives, violating physical symmetry. Our approach PRISM is a unified solution: ReSymNet aligns the rewards temporally, while SymReg acts as the necessary inductive bias to enforce physical reflectional symmetry during this reward shaping.

    We also listed many important papers in the machine learning history that proposed more than one contribution in one paper. It is unfair to reject our paper on this ground.

2. **New Experiments:** Following reviewers’ comments, we conducted all requested experiments during the rebuttal phase.

3. **New Theory:** Following reviewers’ comments, we made all requested extensions to our theory:

    3.1. We generalised our results on the generalisability to reflect the exact structure of our model architecture.

    3.2. We extended our theoretical guarantees from scalar to vector-valued returns.

Best regards,
the authors

---

### Meta-Review · Area_Chair_m1mQ · 2026-01-06

**Summary:**

The submission proposes PRISM to address objective overshadowing in heterogeneous MORL by combining residual-based reward shaping with symmetry regularization. I acknowledge the authors' extensive rebuttal, particularly the inclusion of the GPI-PD backbone and the theoretical extension to vector-valued returns. Empirically the method performs impressively on MuJoCo benchmarks and often exceeds the oracle. However, the decision to reject stems from two fundamental limitations that persist despite the rebuttal. First, the methodological cohesion is weak. ReSymNet and SymReg operate orthogonally and the proposed synergy feels like a concatenation of independent techniques rather than a unified algorithmic contribution. Second, the reliance on manually hardcoded symmetry indices for each environment constitutes a significant dependency on manual feature engineering. This restricts scalability and aligns poorly with the conference focus on learning generalizable representations. The artificial nature of the experimental setup involving masked dense rewards further raises questions about efficacy in naturally sparse settings.

**Reviewer Concerns:**

The rebuttal successfully addressed the theoretical limitations by extending the generalization bounds to vector-valued returns. This resolves the inconsistency between the initial single-objective analysis and the multi-objective setting. The empirical evaluation is now more robust following the inclusion of the GPI-PD backbone and the additional experiments on inverted sparsity settings. These results demonstrate that the method is stable across different architectures and reward structures. The authors also provided sufficient clarifications regarding the random policy initialization and the statistical significance of their results.
Critical concerns remain outstanding regarding the methodology. The connection between the reward shaping and symmetry regularization components is weak. The link between reward heterogeneity and physical asymmetry appears to be a narrative justification rather than a structural requirement of the algorithm. This suggests the method is an aggregation of two orthogonal techniques rather than a unified contribution. The reliance on manually hardcoded symmetry indices for each environment is a significant limitation on generality and constitutes manual feature engineering.

**Reviewer Scores:**

Reviewer s5Dh (Score: 2): The fundamental critique regarding the "condensed papers" feel and lack of a unified storyline remains valid.
Reviewer 41HW (Score: 4): While technical clarifications were provided, the reviewer remained skeptical about the complementarity of the contributions.
Reviewer itAC (Score: 2 -> 4): Might improve slightly due to the responsiveness on new experiments, but the manual decomposition issue remains a blocker for a clear acceptance.
Reviewer pWip (Score: 4): Concerns about the method's generality beyond specific robotic domains persist.

---

### Decision · Program_Chairs · 2026-01-26

Reject